# Ferroelectric ultraviolet photodetector material with ultrafast response speed

Xuexi Yan[1,7], Tingting Yan[2,7], Lingli Li[1], Yi Cao[1], Xinwei Wang[1], Jinghui Wang[1], Ang Tao[1], Tingting Yao[1], Yixiao Jiang [1], Weijin Hu [1], Xiaosheng Fang [2], Hengqiang Ye[3], Xiu-Liang Ma [4,5,6] ✉ & Chunlin Chen [1] ✉

Ferroelectric films are promising to be used for high-performance photodetectors due to their intrinsic electric fields and high dielectric constants. However, the presence of high-density domains with varying polarization directions can severely degrade comprehensive performance. Here, we fabricate high-quality $SrAl_{11-\delta}TiO_{19}$ (SATO) ferroelectric films through a solid-state reaction. The SATO film possesses a magnetoplumbite-type structure with polarization along the c-axis and exhibits the possibility of single-domain ferroelectrics. Ferroelectric performance tests show that the remnant polarization of SATO film reaches 7.8 $\mu C/cm^2$ and the polarization retention exceeds 500 hours. Optoelectronic performance measurements reveal that the SATO photodetector exhibits excellent performance with response wavelength of 330 nm, responsivity of 860 mA/W, detectivity of $1.63 \times 10^{13}$ Jones, switching ratio of $1.9 \times 10^4$, and ultrafast rise/fall response speed of 6.8 ns/17.7 ns (*i.e.*, nearly 10000 times faster than traditional photodetectors). The outstanding properties highlight SATO as an outstanding candidate for next-generation photodetectors.

Photodetectors (PDs) are widely utilized in various fields, including information communications, environmental monitoring, space exploration, etc., owing to their capability to convert optical signals into electrical signals[1–6]. The rapid development of real-time communication, high-precision detection, and deep ultraviolet (UV) information transmission requires PDs to possess ultra-fast response speed, high responsivity, high detectivity, and short response wavelength (< 400 nm)[5]. In traditional PDs, these four parameters are difficult to achieve optimally at the same time. For example, β-$Ga_2O_3$-based p−n junction PDs can obtain high responsivity (>1 A/W) and high detectivity (>$10^{15}$ Jones), but the response speed is very slow (>1 s)[7]. Si-based Schottky PDs have a high response speed of 30 ps and a high efficiency of more than 50%, but they can only detect visible and infrared light due to the small bandgap[8]. AlN-based PDs (~ 6.2 eV) can achieve ms-level deep UV light signal detection, but the responsivity is very low ($10^{-3}$ A/W) due to the very large electrical resistance[9].

Built-in electric field is one of the key factors that tailor the performance of PDs since it affects significantly the separation efficiency of photogenerated carriers[3]. In traditional p-n junction and Schottky PDs, heterointerfaces must be constructed to generate the built-in electric field, which increases the complexity of devices. Moreover, defects such as dislocations, vacancies, and impurities are easily involved in the heterointerfaces, thereby significantly degrading the performance of PDs[10].

Ferroelectric materials have recently been considered as one of the most ideal PD materials because they possess an intrinsic built-in

[1]Shenyang National Laboratory for Materials Science, Institute of Metal Research, Chinese Academy of Sciences, School of Material Science and Engineering, University of Science and Technology of China, 110016 Shenyang, China. [2]College of Smart Materials and Future Energy, State Key Laboratory of Molecular Engineering of Polymers, Fudan University, 200433 Shanghai, China. [3]Ji Hua Laboratory, 528200 Foshan, China. [4]Bay Area Center for Electron Microscopy, Songshan Lake Materials Laboratory, 523808 Dongguan, China. [5]Institute of Physics, Chinese Academy of Sciences, 100190 Beijing, China. [6]State Key Lab of Advanced Processing and Recycling on Non-ferrous Metals, Lanzhou University of Technology, 730050 Lanzhou, China. [7]These authors contributed equally: Xuexi Yan, Tingting Yan. ✉e-mail: xlma@iphy.ac.cn; clchen@imr.ac.cn

electric field induced by the spontaneous ferroelectric polarization, which endows ferroelectric PDs with the advantages of simple structure and stable performance[11,12]. Furthermore, ferroelectric materials can greatly reduce the noise level of PDs since they usually have high dielectric constants[13]. For example, $BaTiO_3$ and $LaFeO_3$ based ferroelectric PDs have achieved the noise level of nA[14,15]. However, current ferroelectric PDs are also facing limitations in their comprehensive performance. For example, although $BiFeO_3$ films achieve ns-level response time, their responsivity remains as low as $10^{-2}$ mA/W[15,16]. $BaTiO_3$/ZnO junction PDs exhibit a high responsivity of up to 100 mA/W but suffer from s-level response time[17]. $\alpha$-$In_2Se_3$ PDs offer a balanced performance in terms of responsivity and response speed but have a low photoelectric conversion efficiency[18]. As well known, most of ferroelectric materials have a high density of ferroelectric domains with different polarization directions as shown in Supplementary Fig. S1, which will not only slow down the response speed of PDs due to the scattering and annihilation of photogenerated carriers at domain walls and the increase in carrier transport distance, but also decrease the responsivity and detectivity of PDs due to the decrease in photocurrent caused by the mutual cancellation of built-in electric fields[19]. To improve the comprehensive performance of ferroelectric PDs, preparing single-domain ferroelectric thin films with large remnant polarization and a wide band gap is a very promising approach, but it still remains unexplored.

$AB_{12}O_{19}$ magnetoplumbite compounds have attracted intense interest due to their abundant physical properties. For example, $SrFe_{12}O_{19}$ is widely used in microwave absorbers, magnetic recording media and sensors, high-frequency electromagnetic (EM) devices, etc., due to its superior magnetic parameters, high magnetic permeability, and low electrical conduction loss[20]. $BaFe_{12}O_{19}$ is the best choice for commercial low-cost permanent magnet materials due to its high coercivity and good chemical and thermal stability. It is widely used in motors, speakers, sensors, and other devices[21]. In addition, $PbFe_{12}O_{19}$ is a natural multiferroic material with broad application prospects in non-volatile memory, ferroelectric photovoltaics, and other fields[22,23]. As a structurally similar derivative, $SrAl_{12}O_{19}$ was predicted to have ferroelectricity but has not yet been experimentally proved[24]. As shown in the schematic atomic model in Supplementary Fig. S2, the unit cell of $SrAl_{12}O_{19}$ comprises alternating rock-salt (i.e., denoted by R) and spinel (i.e., denoted by S) blocks in the sequence of SRS*R* (* denotes 180° rotation around $c$ axis). The misalignment of the positive and negative charge centers of the $AlO_5$ bipyramid in spinel layers makes $SrAl_{12}O_{19}$ exhibit the ferroelectricity. Due to the strong $c$-axis anisotropy, $SrAl_{12}O_{19}$ and related derivatives are good candidates for the preparation of single-domain ferroelectric thin films with polarization direction along the $c$-axis.

In this study, a new ferroelectric compound $SrAl_{11-\delta}TiO_{19}$ (SATO) derived from $SrAl_{12}O_{19}$ was synthesized by the solid-state reaction of AlN and $SrTiO_3$ (STO). The atomic structure of SATO was determined by aberration-corrected transmission electron microscopy (TEM). Measurements of the ferroelectric and optoelectronic properties of SATO thin films were carried out. It was found that the SATO thin films had high remnant polarization ($P_r$) and robust ferroelectric retention. The SATO PDs exhibited outstanding comprehensive performance with ultrafast response speed, high responsivity, high detectivity, low noise, and large switching ratio.

## Results and discussion

Single-crystalline AlN thin films were grown on STO substrates by pulsed laser deposition. High-resolution X-ray diffraction (HRXRD) pattern in Supplementary Fig. S3a suggests that the AlN (0001) thin film and STO (111) substrate have a perfect expitaxial relationship. By annealing the as-prepared AlN thin films in air at 1500 °C for 5 h, SATO thin films were fabricated via the solid-state reaction of AlN and STO. The corresponding HRXRD pattern in Supplementary Fig. S3b

suggests that the SATO has the same magnetoplumbite-type crystal structure as $SrAl_{12}O_{19}$ and grows epitaxially on the STO substrate. To understand the stoichiometry and charge balance of SATO, electron energy-loss spectroscopy (EELS) analysis was carried out. Ti $L_{2,3}$ edges of SATO and STO are shown in Supplementary Fig. S4. It is clear that the Ti $L_{2,3}$ edges of SATO and STO have very similar fine structures with four peaks, suggesting that the Ti ions in SATO have the valence state of +4. To maintain charge neutrality, Al vacancies must exist in SATO. The chemical formula of SATO can be defined as $SrAl_{11-\delta}TiO_{19}$, where $\delta$ represents a parameter related to Al vacancies. Energy dispersive X-ray spectroscopy (EDS) measurements were carried out, and the results are shown in Supplementary Fig. S5 and Supplementary Table S1. After background subtraction, normalization, and integration of the EDS spectrum, the average atomic ratios of Sr, Al, Ti and O in the SATO film are 1.00:10.71:1.02:19.02. The atomic ratio of Sr to Ti is close to 1, similar to that in the $SrTiO_3$ substrate. Since the structure of SATO is derived from $SrAl_{12}O_{19}$ after one Ti atom replaces one Al atom, vacancies must exist in the other eleven sites of Al. Since the valence states of Ti and Al ions are +4 and +3, respectively, the chemical composition of SATO should be $SrAl_{10.67}TiO_{19}$ to reach the complete electrical neutrality ($\delta = 0.33$ in $SrAl_{11-\delta}TiO_{19}$), which is consistent with the EDS measurements considering the accuracy of this technique.

To investigate the microstructure of the SATO thin film and mechanisms for the solid-state reaction of AlN and STO, TEM observations were carried out. Figures 1a and b display the cross-sectional bright-field TEM image and corresponding selected-area electron diffraction (SAED) pattern of the as-deposited AlN thin film. The film-substrate interface is flat and has no secondary phase. The bright/dark contrast within the AlN thin film signifies the columnar growth characteristics. The SAED pattern suggests that the AlN thin film grows epitaxially on the STO substrate with an orientation relationship of AlN (0006)[$11\bar{2}0$] // STO (111)[$\bar{1}10$]. Figures 1c and d show the cross-sectional bright-field TEM image and corresponding SAED pattern of the SATO thin film. As one can see, both the film-substrate interface and film surface are flat. The SATO thin film has a similar thickness to the AlN thin film. The epitaxial relationship between SATO thin film and STO substrate is SATO (0001)[$11\bar{2}0$] // STO (111)[$\bar{1}10$]. These results suggest that Sr and Ti atoms in the STO substrate diffused into the AlN thin film, then the solid reaction occurred in air and formed the SATO thin film. In addition, it is very clear that the SATO thin film exhibits a uniform contrast and has no visible ferroelectric domain walls, indicating that the as-prepared SATO thin film exhibits the possibility of single-domain ferroelectrics. SAED tilting series of SATO are shown in Supplementary Fig. S6, from which we confirmed that the SATO has a hexagonal magnetoplumbite crystal structure with lattice constant of a = b = 5.74 Å, c = 22.32 Å, $\alpha = \beta = 90°$, and $\gamma = 120°$. The bright-field TEM images of the SATO thin film along the [$10\bar{1}0$] and [0001] zone axes are shown in Supplementary Fig. S7. There are no ferroelectric domain walls that can be observed in the SATO thin film, which indicates that the SATO ferroelectric film probably has a single-domain structure.

To reveal the atomic structure of SATO, high-angle annular dark-field (HAADF) and annular bright-field (ABF) imaging were carried out in a scanning transmission electron microscope (STEM). HAADF images can show heavy atomic columns since the intensity of atomic columns is approximately proportional to $Z^{1.7}$ (Z: atomic number)[25], while the ABF images can present all the atomic columns, including O[26]. Figure 2 shows the HAADF and ABF STEM images of SATO along the [$11\bar{2}0$], [$1\bar{1}00$], and [0001] zone axes. Atomic models are inserted into each image for a better understanding of the atomic structure of SATO. As shown in Fig. 2a and d, the SATO consists of alternating rock-salt and spinel blocks in the sequence of SRS*R* along the $c$-axis. Sr atoms occupy the center of O dodecahedrons in the rock-salt blocks. Al atoms occupy five distinct Wyckoff positions (Please refer to Fig. S2): 2a, 12k, 4f$_1$ in O octahedra, 4f$_2$ in O tetrahedra, and 2b in O

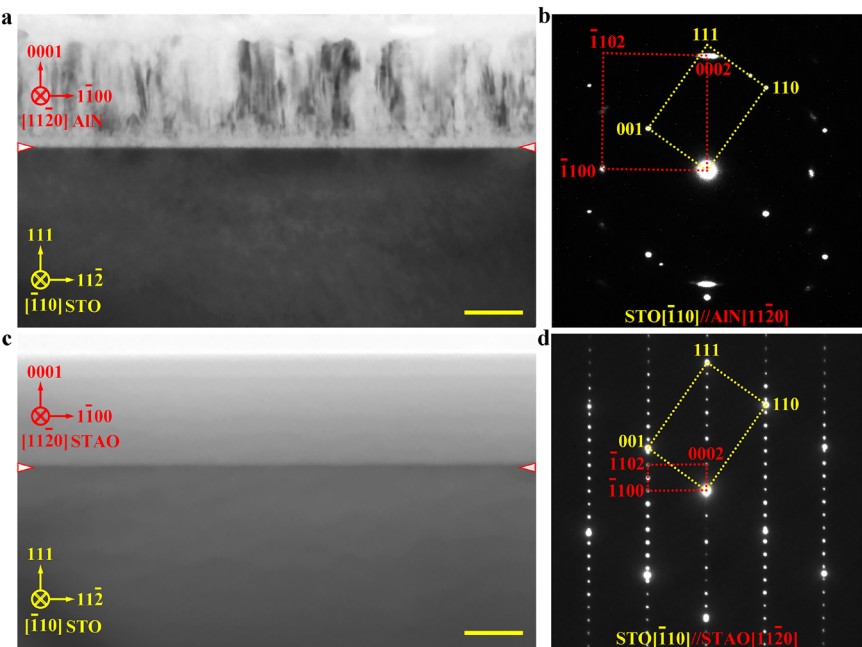

**Fig. 1 | Microstructure of the AlN and SATO thin films on STO (111) substrates.** **a** Cross-sectional TEM image and **b** Corresponding SAED pattern of the AlN film along the STO [$\bar{1}$10] // AlN [11$\bar{2}$0] zone axes. The epitaxial AlN thin film exhibits columnar growth characteristics. **c** Bright-field TEM image and **d** Corresponding SAED pattern of the SATO thin film formed by annealing of the AlN/STO thin film at high temperature in air. The SATO thin film exhibits uniform contrast due to its high quality. Scale bar, 20 nm.

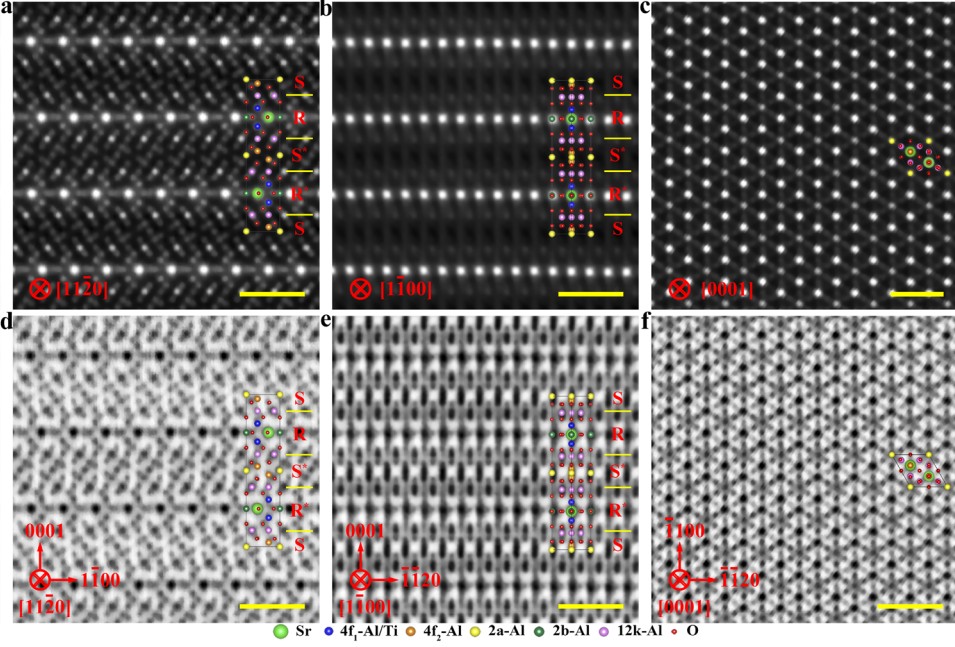

● Sr  ● 4f₁-Al/Ti  ● 4f₂-Al  ● 2a-Al  ● 2b-Al  ● 12k-Al  ● O

**Fig. 2 | Atomic structure of the SATO thin film observed along the three characteristic low-index zone axes.** **a**–**c** HAADF STEM images along the [11$\bar{2}$0], [1$\bar{1}$00], and [0001] zone axes of the SATO thin film, respectively. **d**–**f** Corresponding ABF STEM images. The atomic models were attached to each image. The green balls represent Sr atoms. The orange, yellow, pink, and dark green balls represent Al atoms. The blue balls represent Ti or Al atoms occupying the same symmetry position. The red balls represent O atoms. Scale bar, 1 nm.

bipyramids. All the HAADF and ABF images in Fig. 2 confirm that the SATO has a hexagonal magnetoplumbite crystal structure like SrAl₁₂O₁₉. To determine the atomic sites of Ti in SATO, atomic-resolution EDS elemental maps along the [11$\bar{2}$0] zone axis were obtained and shown in Fig. 3. As indicated by white arrows, the distribution of Ti atoms in SATO is ordered, and they partially substitute the Al atoms at the 4f₁ Wyckoff position. Simulated HAADF and ABF images of SATO along the [11$\bar{2}$0], [1$\bar{1}$00], and [0001] zone axes using the atomic model of hexagonal magnetoplumbite are shown in Supplementary Fig. S8. They are consistent well with the experimental images shown in Fig. 2, which proves again the hexagonal magnetoplumbite crystal structure of SATO.

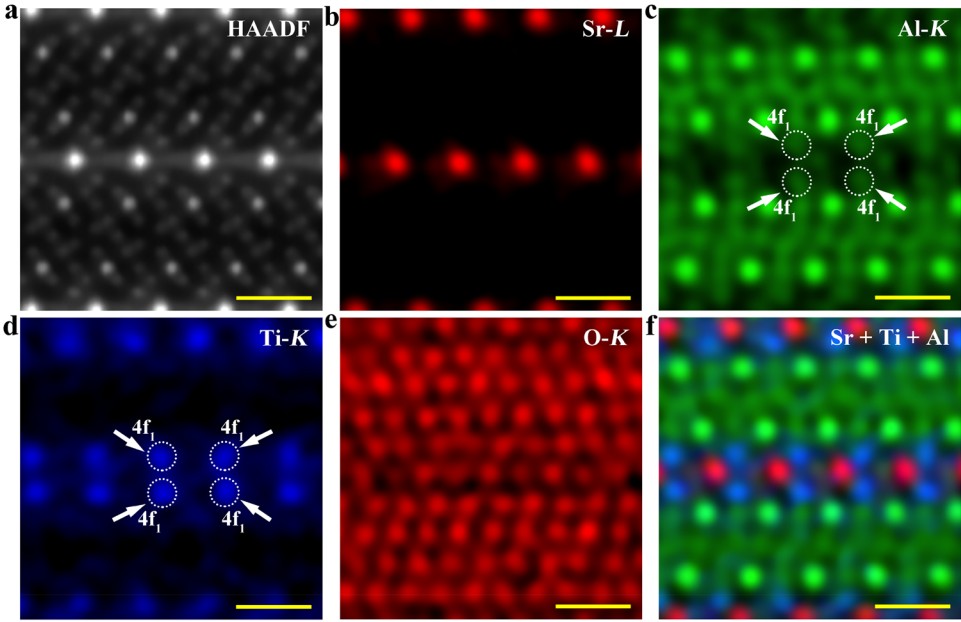

**Fig. 3 | Atomic-resolution elemental mapping of the SATO thin film. a** HAADF STEM image of the SATO thin film. **b–e** Corresponding Sr-*L*, Ti-*K*, Al-*K*, and O-*K* maps. **f** The overlay maps of the Sr, Ti, and Al. Ti atoms partially replaced Al atoms of $4f_1$ Wyckoff positions, as indicated by the arrows. Scale bar, 5 Å.

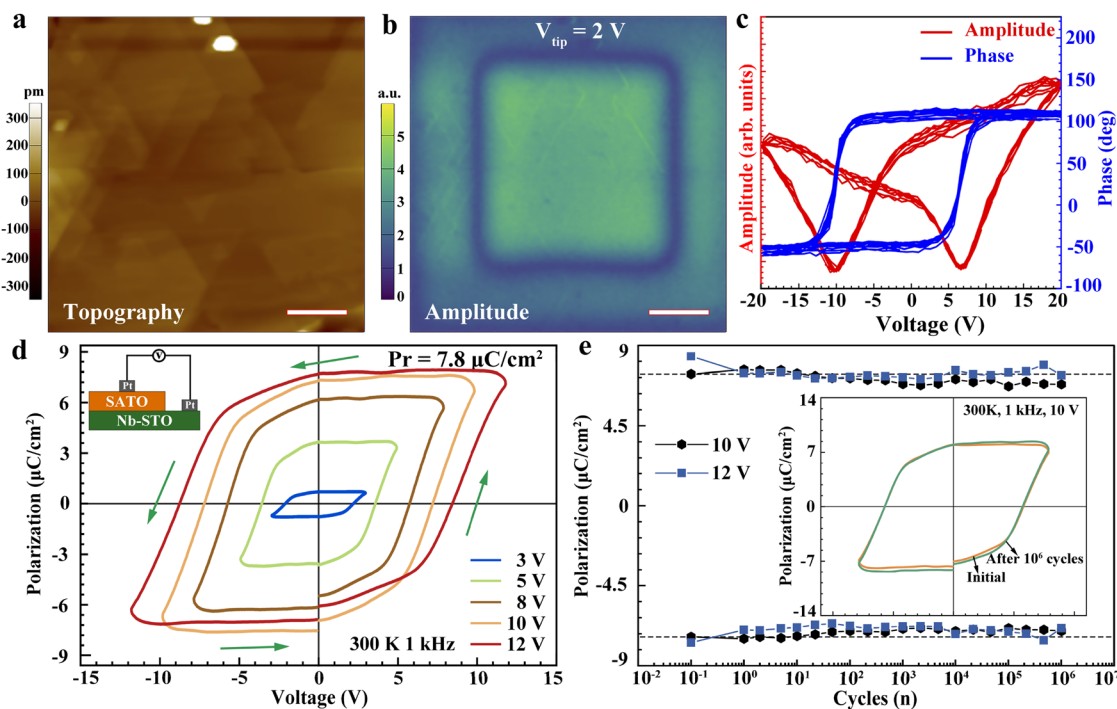

**Fig. 4 | Ferroelectric and piezoelectric properties of the SATO thin film. a** PFM topographic image of the SATO thin film. The RA is about 2 nm. **b** PFM amplitude image polarized by ± 10 V bias voltages. The voltage of the probe tip ($V_{tip}$) was 2 V. **c** The local PFM amplitude (red) and phase (blue) hysteresis loops. **d** The polarization-electric field (P-E) hysteresis loops of the SATO thin film under different voltages at 300 K, 1 kHz. The SATO thin film has a remanent polarization of 7.8 μC/cm² and exhibits typical intrinsic ferroelectric characteristics. **e** Fatigue characteristics of the SATO thin film under 10 V and 12 V. Inset are the PUND loops before and after cycling. After $10^6$ cycles, the polarization intensity exhibits a strong fatigue stability without significant attenuation. Scale bar of (**a**) and (**b**), 2 μm.

Ferroelectric and piezoelectric properties of the SATO thin film were measured at room temperature, and the results are shown in Fig. 4. Figure 4a presents the surface morphology of the SATO thin film obtained by piezoresponse force microscopy (PFM). The SATO thin film displays a well-defined hexagonal plate-like structure with a surface roughness (RA) of ~2 nm, indicating high-quality film growth. Figure 4b elucidates the piezoelectric properties through PFM lithography and local switching spectroscopy (SS). The local ferroelectric switching behavior characterized by phase-voltage hysteresis and amplitude-voltage butterfly loops in Fig. 4c can preliminarily prove

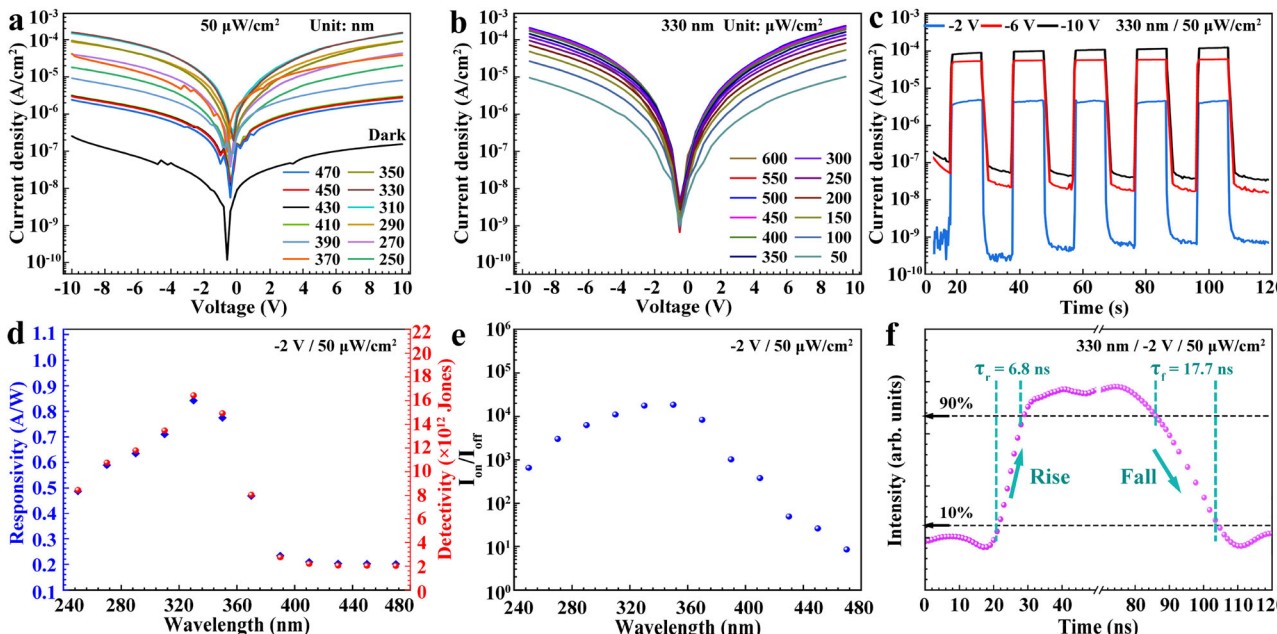

**Fig. 5 | Photoelectric properties the SATO films. a** Logarithmic I-V curves of the SATO PDs under different light wavelengths (250 - 470 nm). The SATO PDs have the strongest response to 330 nm UV light. **b** I–V curves of the SATO PDs under different power densities at 330 nm. **c** I–T curves under 330 nm on-off illumination at −2, −6, and −10 V of the SATO PDs. **d** Responsivity, detectivity, and **e** On/off ratio curves at different wavelengths. At 330 nm, the responsivity, detectivity, and on/off ratio of the SATO PDs at −2 V are as high as 860 mA/W, $1.63 \times 10^{13}$ Jones, and $1.9 \times 10^{4}$, respectively. **f** Time-resolved transient photoresponse curves. The SATO PDs exhibit fast response speed with a rise time ($\tau_r$) of 6.8 ns and a fall time ($\tau_f$) of 17.7 ns.

that SATO thin films may have ferroelectricity. To determine the initial domain state of the virgin SATO film and verify the reproducibility of the PFM measurements, we performed multiple measurements at four randomly selected positions on a $10 \times 10$ mm sample under nearly identical measurement conditions, as shown in Supplementary Fig. S9. The initial piezoelectric amplitude and phase images at all four positions display uniform contrast, which indicates that the virgin SATO film possibly possesses a single-domain nature. By applying a custom pattern with +10 V and −10 V DC voltages, the large-scale amplitude and phase images reveal distinct 180° contrast polarization signals between the center and edge regions, highlighting the robust piezoelectric response of the SATO thin film. The PFM measurements at all positions showed similar characteristics, indicating that the SATO film is uniform with a good reproducibility of ferroelectric properties. Remarkably, as one can see from the Supplementary Fig. S10, ferroelectric retention measurements demonstrate that the ferroelectric polarization in the SATO thin film can be maintained for over 500 h without significant degradation, showcasing an excellent long-term stability. To exclude the possibility that the ferroelectricity comes from the AlN thin film grown on the Nb:STO substrate, we performed PFM characterizations and showed the results in Supplementary Fig. S11. As can be seen, it is difficult to write domains in the AlN film, which demonstrates that the AlN thin film is not ferroelectric.

To unequivocally establish the intrinsic ferroelectric nature of the SATO thin film, the macroscopic polarization-electric field (P-E) hysteresis loops were measured through the Positive-up-negative-down (PUND) method under different voltages at 300 K and 1 kHz, as shown in Fig. 4d. The PUND mode is one of the most accurate methods for measuring the remanent polarization of ferroelectric materials. It can eliminate the influence of dielectric and leakage effects and reflect the intrinsic characteristics of ferroelectricity. Supplementary Fig. S12 shows the measurement parameters of the PUND mode. In the test, the write voltage and read voltage were both 10 V. It is clear that the remanent polarization gradually increases with the voltage. After the voltage is increased to 10 V and higher, the remanent polarization

remains unchanged ($\sim 7.8 \, \mu C/cm^2$), and the polarization saturation is reached, as shown in Fig. 4d. The coercive field ($E_c$) of the SATO thin film is up to 21 MV/m at 10 V voltage. These ferroelectric properties are comparable to those of $BiFeO_3$[27], indicating that SATO is a promising ferroelectric material. Figure 4e shows the results of ferroelectric fatigue testing. After $10^6$ cycles, the remanent polarization intensity did not attenuate significantly, which indicates that the SATO film has a strong fatigue stability. The atomic origin of the ferroelectricity of SATO has been experimentally identified as the displacement of Al atoms within the $AlO_5$ bipyramids, as depicted in Supplementary Fig. S13 and Fig. S14. As can be seen, the polarization of SATO is vertically upward, indicating that SATO is a uniaxial ferroelectric along the c-axis. Thus, the SATO thin film belongs to the first-type displacement-type ferroelectrics.

To reveal the performance of ferroelectric SATO film as a photodetector, we carried out photoelectric response tests. Supplementary Fig. S15 shows the schematic diagram of SATO photodetector testing. Due to the existence of spontaneous polarization in the SATO thin film, a built-in electric field is generated in the opposite direction of the polarization. When high-energy light irradiates the ferroelectric thin film, photogenerated carriers are quickly separated due to the built-in electric field, thereby converting the received optical signal into an electrical signal for detection. To elucidate the optoelectronic properties of the SATO thin film, we conducted measurements using a comprehensive photoelectric testing system equipped with a monochromator, and showed the results in Fig. 5. Figure 5a shows the logarithmic I-V curves of the SATO thin film under different light wavelengths ranging from 250 to 470 nm. In the dark, the SATO thin film exhibits a low current density of less than 1 nA/cm², indicating a very low noise level when used as a PD material. Under illumination, the photocurrent of the SATO PDs significantly increases, demonstrating effective separation of photogenerated carriers within the thin film. The photocurrent reaches its maximum at a wavelength of 330 nm, suggesting that SATO is highly suitable for fabricating UVB (280–320 nm) PDs. In addition, Supplementary Fig. S16 shows that the

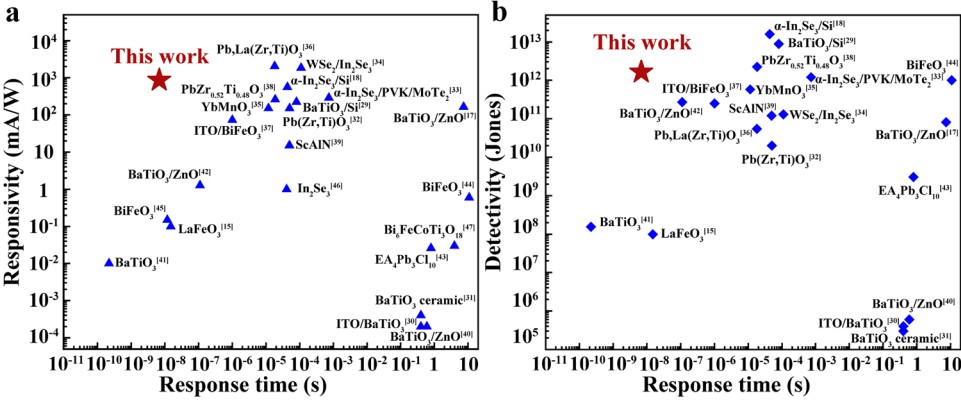

**Fig. 6 | Comparison of the properties of ferroelectric photodetectors. a** Responsivity and (**b**) detectivity of ferroelectric PDs. The SATO PDs exhibit outstanding comprehensive properties with high responsivity, high detectivity, and ultrafast response speed.

SATO film has a large dielectric constant ($>16$), low dielectric loss ($<0.1$), and excellent frequency stability at room temperature. Figure 5b displays the I-V curves under different light intensities at 330 nm. The SATO thin film shows excellent detection capability for both weak light (down to 50 μW/cm²) and strong light (up to 500 μW/cm²), indicating its proficiency in detecting weak signals. Figure 5c presents the I-T curves of the SATO PDs at 330 nm under bias voltages of −2, −6, and −10 V. The photocurrent density increases with increasing bias voltage, reaching a maximum of approximately $10^{-4}$ A/cm² at −10 V. At a bias of −2 V, the SATO detector exhibits the largest difference between photocurrent and dark current, implying the best signal-to-noise ratio for information detection. Furthermore, long-term repeated on-off switching measurements shown in Supplementary Fig. S17 reveal that the photocurrent and dark current remain stable, and the SATO detector has excellent reproducibility and detection reliability.

To quantitatively evaluate the performance of the SATO PDs, we prepared three devices with different effective areas (Supplementary Fig. S18) to calculated important parameters such as responsivity (R), detectivity (D*), and on/off ratio ($I_{on}/I_{off}$) using the following formulas:

$$R = \frac{I_{ph}}{PS} \tag{1}$$

$$D^* = \frac{R}{\sqrt{2qJ_d}} \tag{2}$$

$$\frac{I_{on}}{I_{off}} = \frac{I_{ph}}{I_d} \tag{3}$$

where $I_{ph}$, P, S, q, $J_d$, and $I_d$ represent the photocurrent, light intensity, effective device area, electronic charge, dark current density, and dark current, respectively[28]. As depicted in Fig. 5d, the responsivity and detectivity of the SATO PDs increase with the wavelength from 250 nm to 320 nm. Beyond 320 nm, both parameters sharply decrease with the increasing wavelength. At 320 nm under a bias of −2 V, the responsivity and detectivity of the SATO film reach as high as 860 mA/W and $1.63 \times 10^{13}$ Jones, respectively, which significantly outperform most currently known ferroelectric materials. Supplementary Fig. S19 shows the responsivity and detectivity of different devices at different laser power intensities under 330 nm illumination. The three devices exhibit the same feature. The responsivity and detectivity gradually decrease with the increase of laser intensity. Figure 5e shows the on/off ratios at different wavelengths under a bias of −2 V, with the highest ratio exceeding $10^4$ at 320 nm. To reveal the response speed of the SATO

PDs, a time-resolved photoresponse test was conducted, as shown in Fig. 5f. Generally, the response speed is the time taken for the photocurrent to transform from 10% to 90%. As can be seen, the rise time ($\tau_r$) and fall time ($\tau_f$) took only 6.8 ns and 17.7 ns respectively, which is nearly 10000 times faster than traditional photodetectors like BaTiO₃[29–31] and PbTiO₃[32]. The photoelectric switching response and transient photoelectric response of SATO with different polarization directions were measured. As shown in Supplementary Fig. S20, the SATO photodetector has a strong and stable photoelectric response regardless of positive or negative polarization. More importantly, as shown in Supplementary Fig. S21, the SATO photodetectors with positive and negative polarizations have ultra-fast response speed. Their rise/fall times are 6.6 ns/17.7 ns and 8.7 ns/12.2 ns, respectively. Similar tests using other devices with different effective areas were carried out. The current intensity, effective area, laser intensity, responsivity, detectivity and response time of three SATO photodetector devices are listed in the Supplementary Table S2. All the SATO photodetector devices exhibit fast response speed. The ultrafast (ns-scale) response speed indicates that SATO-based optoelectronic devices can be applied to many fields, such as instant communication, which represents an important progress in the field of PD materials.

To compare the comprehensive performance of the SATO ferroelectric PDs with other ferroelectric PDs, Fig. 6 present the response times, responsivities, and detectivities of the ferroelectric PDs reported so far[15,17,18,29–47]. It is clear that most ferroelectric PDs have response times ranging from seconds to milliseconds; only a few achieve the microsecond level. The responsivities and detectivities are generally less than 200 mA/W and $10^{12}$ Jones, respectively. Although some devices achieve response speeds of the nanosecond level, their responsivity and detectivity are far below 0.01 mA/W and $10^8$ Jones, which do not meet the requirements for PD applications. As one can see, the SATO ferroelectric PDs in this work have not only ultrafast response speed ($<10$ ns), but also very high responsivity ($>500$ mA/W) and detectivity ($>10^{13}$ Jones). Thus, the SATO ferroelectric PDs exhibit great potential in applications for real-time detection and the optoelectronic communication industries.

In summary, we have successfully developed a SATO ferroelectric PD thin film through a solid-state reaction of AlN thin film and STO substrate. The SATO thin film exhibits a magnetoplumbite-type uniaxial ferroelectric structure with a remnant polarization up to 7.8 μC/cm² and robust ferroelectric retention exceeding 500 h. Optoelectronic performance measurements reveal that the SATO thin film is an outstanding ultraviolet photodetector material with a response wavelength of 330 nm, responsivity of 860 mA/W, detectivity of $1.63 \times 10^{13}$ Jones, switching ratio of $1.9 \times 10^4$. More importantly, the SATO photodetector has an ultrafast response speed with a rise time of

6.8 ns and a fall time of 17.7 ns. The superior performance, driven by the displacement of Al within oxygen trigonal bipyramids, positions SATO as a promising candidate for advanced photodetection applications in many fields, such as real-time communication, space exploration, pharmaceuticals, environmental monitoring, and so on.

## Methods

### Materials and microscopic observations

Epitaxial AlN thin films were grown on STO (111) substrates ($10 \times 10 \times 0.5$ mm) using the pulsed laser deposition (PLD, KrF excimer laser, 248 nm) method[48–50]. Before growth, the STO substrates were annealed in air at 1000 °C for 10 h to obtain an atomically flat surface, and then transferred to a high vacuum chamber ($< 10^{-5}$ Pa) and baked at 800 °C for 1 h. For the growth of AlN thin films, the growth temperature of 850 °C, $N_2$ pressure of $10^{-3}$ Pa were adopted. The energy density of 3.5 J/cm$^2$ and frequency of 5 Hz of the laser were used. The distance between the target and the substrate was set as 8 cm. After deposition, the AlN films were annealed at 1500 °C in air for 5 h. As a result, single-crystal epitaxial SATO thin films were formed by the solid-state reaction of AlN thin films and STO substrates.

TEM samples were prepared using the standard procedures of ion-milling, including cutting, mechanical grinding, dimpling, and Ar-ion thinning. To obtain electron-transparent and minimally damaged samples, the thinning voltage was progressively reduced from 5 keV to 0.1 keV, and the thinning angle was adjusted from 7° to 5°. Bright-field imaging and SAED patterns were recorded at 200 kV using a JEM-2100 (JEOL Co., Ltd) TEM. STEM HAADF, ABF images, and the atomic-scale elemental mapping were obtained using an aberration-corrected STEM (Themis 300, FEI) equipped with a probe corrector (CEOS, GmbH) and super-EDS. The acceleration voltage, convergence angle, and camera length were set to 300 kV, 21.4 mrad, and 115 mm, respectively. Collection semiangles for HAADF and ABF were 50 - 200 mrad and 11 – 22 mrad, respectively. HAADF and ABF images were simulated based on the multi-slice theory using the Dr.Probe software package[51,52]. This approach allowed for accurate modeling of the complex interactions between the electron beam and the sample, providing insights into the structural details observed in the experimental TEM images. The simulation parameters were adopted according to the experimental conditions.

### Ferroelectric and PD performance measurements

The electrodes were fabricated using a photolithography technique, followed by sputtering the Pt top electrode with a thickness of 50 nm and a size of $50 \times 50$ μm$^2$ on the film[53]. The polarization-electric field (P–E) hysteresis loop was measured on a ferroelectric tester (Precision Multiferroic, Radiant Tech, USA.; TF-Analyser 3000, aixACCT, Germany) at 1 kHz and room temperature (25 °C) employing the Remanent Hysteresis or Positive-up-negative-down (PUND) mode. Piezoresponse force microscopy (PFM) measurements were performed using a Cypher Asylum Research platform to characterize the local piezoelectric response. Conductive Pt-coated silicon cantilevers were used, and the sample was mounted on the PFM stage by conductive adhesive tape. Measurements were conducted in contact mode, with the cantilever in contact with the sample surface and a 10 V, 20 kHz AC voltage applied between the tip and the bottom electrode. The out-of-plane piezoresponse signal was recorded as a function of the applied voltage, and the PFM amplitude and phase images were obtained to visualize the ferroelectric domain structure.

The optoelectronic performance was measured using a Keithley 4200 semiconductor characterization system and a xenon (Xe) lamp equipped with a monochromator, providing a tunable light source with wavelengths ranging from 200 nm to 900 nm[54]. For transient photoconductivity measurements, a sophisticated transient photoresponse system was employed. This system comprised a neodymium-doped yttrium aluminum garnet (Nd:YAG) laser (Continuum Electro-Optics, MINILITE II) with a pulse duration of 2 – 3 ns at 330 nm, a resistor, and an oscilloscope (Tektronix MSO/DPO5000). This setup enabled precise measurement of the sample's response to light pulses, offering comprehensive insights into its optoelectronic properties.

## Data availability

The presented data were available from the corresponding authors upon request.

## Code availability

The computer code that supports the findings of this study is available from the corresponding authors upon request.

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

## Acknowledgments

This study was supported by the National Key Research and Development Program of China (No. 2024YFA1408000 (C.C)), the National Natural Science Foundation of China (Grant Nos. 52125101 (C.C), W2511055 (C.C), 52271015 (T.Yao), 52401026 (X.Y)), the China Post-doctoral Science Foundation (Grant Nos. GZC20232748 (X.Y), 2024M753303 (X.Y), BX20240087 (T.Yan)), Jihua Laboratory (Project No. X210141TL210 (C.C)), the Guangdong Major Project of Basic Research (Grant No. 2021B0301030003 (C.C)). We thank Dr. Jingping Cui, Changji Li, and Yan Liang of the Institute of Metal Research for support in experiments.

## Author contributions

X.Y. performed the experiments of film growth and TEM characterizations, analyzed data, and wrote the paper. T.Yan performed the photoelectric performance test and analyzed data. L.L. and Y.C. performed the ferroelectric performance test. X.W., J.W., A.T., T.Yao, and Y.J. helped to analyze data. W.H. and X.F. helped to direct the performance test. H.Y., X.M., and C.C. directed the entire study. All authors read and commented on the paper.

## Competing interests

The authors declare no competing interests.
