## [Transparent Peer Review file · Nature Communications]

Ferroelectric ultraviolet photodetector material with ultrafast response speed

Corresponding Author: Professor Chunlin Chen

Version 0:

Reviewer comments:

Reviewer #1

(Remarks to the Author)

This manuscript describes the optoelectronic performance of the SATO thin film, which shows a high ultraviolet photodetection sensitivity with a responsivity of 860 mA/W, a detectivity of 1.63×10^{13} Jones, and a switching ratio of 104. The SATO photodetector has an extremely ultrafast response speed with a rise time (τ_r) of 6.8 ns and a fall time (τ_f) of 17.7 ns. Major comments/corrections are as follows for further clarification and improvement.

1) The manuscript highlights the potential of ferroelectric thin films for high-performance photodetectors, primarily due to their intrinsic electric fields and polarization-related effects. To further support this claim, it would be beneficial to include a detailed analysis of the dielectric properties of the SATO thin film. Parameters such as dielectric constant, dielectric loss, and their frequency dependence would provide valuable insight into the material's capability for charge storage and screening, which are critical for optimizing photodetector performance. Including these measurements would strengthen the discussion on the multifunctional role of ferroelectricity in optoelectronic applications.

2) The statement "AB₁₂O₁₉ magnetoplumbite compounds have garnered significant attention owing to their multifunctional physical properties, including ferromagnetism, ferroelectricity, high dielectric constant, and wide bandgap. These characteristics make them promising candidates for applications in spintronics, microwave devices, and multifunctional electronic materials." It is superficial in nature, without any strong evidence/reference.

3) The polarization hysteresis data presented in Fig. 4f convincingly demonstrate the ferroelectric nature of the SATO thin film, with a notable remanent polarization (P_r) of 7.8 $\mu\text{C}/\text{cm}^2$ and a coercive field (E_c) of 21 MV/m at 300 K and 1 kHz. To further substantiate the intrinsic ferroelectric behavior and rule out potential non-ferroelectric contributions (e.g., from leakage or dielectric effects), it would be valuable to include Positive-Up-Negative-Down (PUND) measurements. Additionally, a fatigue analysis showing the stability of polarization over repeated electric field cycling would strengthen the case for the material's potential in non-volatile memory or other ferroelectric device applications. Including these characterizations would significantly enhance the credibility and completeness of the ferroelectric performance assessment. More importantly, the authors have NOT highlighted the primary cause of the development/origin of polarization; it must be illustrated in the revised manuscript

4) The HAADF and ABF STEM images of the SATO thin film along the $[11\bar{2}0]$, $[1\bar{1}00]$, and $[0001]$ zone axes are clearly presented, and the overlaid atomic models aid significantly in visualizing the local atomic arrangement. Polarization mapping based on the HAADF-STEM data is recommended to enhance the structural analysis further. Quantitative mapping of cation displacements—especially the relative shifts between A-site and B-site cations or between cations and anions—could provide direct insight into the local polarization vectors and their directional variation. This addition would confirm the presence of spontaneous polarization at the atomic scale and establish a stronger structure-property correlation for the observed ferroelectric behavior in SATO.

5) In Figure 5, (a) & (b), legends are not showing clearly, so increase the size; (b) all the graphs overlapped at different power intensities, so enlarge this figure; (c) caption does not mention power intensity; (e) Y-axis label doesn't show clearly, so increase the size, (f) not mentioned wavelength and power intensity, so also mention in the caption.

6) The transient photoconductivity measurements and their results make this paper unique and might deserve publication in Nature Communications. However, the authors have just put one graph, Fig. 5 (f), for the same. It requires its repeatability (long time ON/OFF), current intensity (absolute value), different laser intensity, device effective area (at least three sizes of the device), etc., to calculate the responsivity and detectivity of the photodetector (PD) as claimed!! I would be happy to see the detailed analysis of Fig. 5 (f), which supports and corroborates the claims. A table (supplementary) is needed for all the inputs with responsivity and detectivity calculations.

Reviewer #2

(Remarks to the Author)

This manuscript presented SrAl₁₁TiO₁₉ (SATO) thin films as a novel ferroelectric material for ultrafast ultraviolet photodetectors. While the TEM structural analysis is generally well-executed, the ferroelectric characterization requires substantial reinforcement. Several additional issues warrant attention.

Major Concerns

1. Inadequate Ferroelectric Characterization

P-E Hysteresis Analysis: The P-E loops presented in Figure 4f lack essential electric field dependence data. Demonstrating polarization saturation at high fields is critical for confirming intrinsic ferroelectric switching behavior. The current data are insufficient to distinguish between true ferroelectric switching and other polarization mechanisms.

PFM Domain Analysis: While the PFM images successfully demonstrate voltage-induced domain writing (Figures 4b and 4d), the study lacks PFM characterization of pristine, as-grown film regions. This baseline comparison is crucial for determining whether the observed ferroelectricity represents an intrinsic material property or results from field-induced effects.

2. Stoichiometry and Charge Balance Issues

The reported SATO stoichiometry raises fundamental questions about charge neutrality. The manuscript must address how electronic balance is maintained in this composition—specifically, the oxidation state of titanium and the potential role of intrinsic defects. This analysis is crucial for understanding both the insulating properties and ferroelectric behavior of SATO.

3. Substrate Interaction Effects

The SATO films are grown on Nb-doped STO substrates and processed at 1500C. At these elevated temperatures, Nb diffusion from the substrate into the SATO layer is highly probable. The authors must evaluate this possibility and discuss its potential impact on film conductivity, ferroelectric properties, and photoresponse characteristics.

4. Polarization-Dependent Photoresponse

A defining characteristic of ferroelectric photodetectors is their ability to exhibit polarization-switchable performance. The study lacks critical experiments demonstrating how photoresponse parameters (photocurrent magnitude, responsivity, response time) vary with deliberate polarization reversal in the SATO film. Such measurements would provide compelling evidence for ferroelectricity-enhanced photodetection.

Minor Concerns

Device Visualization: Including a photograph of the fabricated SATO photodetector would help readers better understand the device architecture and experimental configuration.

Domain Structure Representation: The schematic diagram of ferroelectric domains in Supplementary Figure S1 does not accurately reflect typical domain patterns observed in ferroelectric thin films. Domain illustrations should be grounded in experimental observations from either the literature or the authors' work to ensure scientific accuracy and educational value.

Addressing these concerns—particularly the ferroelectric characterization gaps and stoichiometry questions—will substantially strengthen the manuscript's scientific rigor and enhance the credibility of claims regarding this novel ferroelectric photodetector material.

Reviewer #3

(Remarks to the Author)

In the paper "Ferroelectric ultraviolet photodetector material with ultrafast response speed", Yan et al present an outstanding optoelectronic performance of the single-domain ferroelectric SATO thin film. The ferroelectric properties have been discussed and they provide excellent optoelectronic performances for ultraviolet region, such as responsibility of 860mA/W, the rise time of 6.8 ns and the fall time of 17.7 ns. I think the paper is well organized, and the presented experimental results are very interesting. However, several important points concerning the ferroelectric properties and the underlying mechanisms that enable such high optoelectronic performance remain unclear.

1. The authors used the PFM and polarization hysteresis loops to show the ferroelectric behavior of SATO thin film. While the authors state that the high dielectric constants of ferroelectric materials lead to lower noise level, it will be helpful if they can show dielectric behavior of SATO thin film and compare with other ferroelectric materials.

2. The authors state that the ferroelectric polarization induces intrinsic built-in electric field increasing optoelectronic

properties. I am wondering whether the ferroelectricity can be tailored in SATO thin film, that the effect of ferroelectric polarization can be confirmed. They could measure optoelectronic performances using the SATO thin films with different spontaneous polarization.

3. It was demonstrated that the single-domain ferroelectric thin films can result the high optoelectronic performances. Is all single-domain ferroelectric thin films are possible to result the high optoelectronic performances? If not, what distinguishes SATO from other candidates? A brief comparison with other single-domain ferroelectric materials would help contextualize the advantages of the SATO system.

4. The SATO-based device demonstrates a nanosecond-scale response time, which is remarkable. What could be origin of the ultrafast response speed compared to other conventional photodetectors?

Version 1:

Reviewer comments:

Reviewer #1

(Remarks to the Author)

The authors have improved the manuscript in line with queries, comments, and concerns raised by reviewers; however, I am still not convinced why these SATO thin films show an increase in coercive field even after reaching of maximum polarization. An ideal ferroelectric system shows nearly the same coercive field after polarization saturation. Polarization graphs also show nearly square hysteresis without saturation (as can be seen PFM data saturation after 12 V); it requires some explanation by the authors.

Reviewer #2

(Remarks to the Author)

The authors have made substantial improvements to the manuscript and addressed most of the concerns raised in the initial review. The additional experimental data, particularly, polarization-dependent photoresponse characterization, significantly strengthen the work. However, two important issues require further attention:

While the authors have provided PFM measurements of the AlN precursor film (Supplementary Fig. S9), this does not address the original concern about the pristine SATO film state. Required additional data is PFM amplitude and phase images of virgin SATO film regions (without any prior electrical poling).

The current response addresses whether the precursor AlN film is ferroelectric, but the fundamental question about the intrinsic domain state of the SATO film remains unanswered.

The authors propose Al vacancies as the charge compensation mechanism for Ti^{4+} substitution. However, there appears to be an inconsistency between this explanation and the reported compositional analysis. The EDS analysis reports Sr:Al:Ti:O = 1:11:1:19, suggesting single Ti substitution. If the charge balance requires Al vacancies (as stated: "Al vacancies must exist in SATO"), the expected stoichiometry would differ from $\text{SrAl}_{11}\text{TiO}_{19}$. A more detailed explanation is needed for how the proposed defect chemistry (Al vacancies + Ti^{4+} substitution) reconciles with the experimental composition data

Reviewer #3

(Remarks to the Author)

We thank the authors for their detailed and comprehensive reply. They provided additional information to clarify their claims. The paper is now much clearer and can be considered for publication.

Version 2:

Reviewer comments:

Reviewer #1

(Remarks to the Author)

The paper may be accepted for publication

Reviewer #2

(Remarks to the Author)

The authors have made substantial improvements to address most of the concerns raised in the initial review, and their efforts are acknowledged. However, the newly added Fig. S9 PFM data introduces a serious contradiction that fundamentally challenges the manuscript's central claims.

The PFM measurements in Fig. S9 demonstrate that the as-prepared SATO sample exhibits characteristics consistent with a paraelectric phase. While the amplitude image is presented in arbitrary units (a.u.), preventing direct quantitative comparison with the results obtained under applied field, the phase image reveals distinctly different values compared to those observed after electrical poling (Fig. 4b, 4d). If the as-prepared sample were truly in a ferroelectric phase, the phase values should show consistency between the virgin state and post-poling measurements, at least within domains of the same polarization orientation.

The collective PFM results strongly suggest the possibility of a field-induced ferroelectric phase transition rather than intrinsic ferroelectricity. However, this interpretation directly contradicts the TEM structural analysis, which concludes that the as-prepared sample possesses a ferroelectric phase based on observed atomic displacements and structural characteristics.

These conflicting interpretations cannot be reconciled without further clarification and additional experimental evidence. Therefore, the manuscript cannot be considered for publication until this critical contradiction is satisfactorily resolved.

Version 3:

Reviewer comments:

Reviewer #2

(Remarks to the Author)

The manuscript reports highly intriguing and potentially impactful experimental results regarding ultrafast ferroelectric photodetectors with exceptional performance characteristics. The main finding of nanosecond-scale response times combined with high responsivity and detectivity appears robust and could attract broad interest within the optoelectronics community. However, the interpretation of these results relies on several supporting datasets, which unfortunately exhibit significant inconsistencies and lack reproducibility.

In particular, the additional PFM data provided in response to the review (Fig. S9) do not reconcile with the results in Fig. 4 and raise new concerns about reproducibility. In Fig. 4, PFM measurements demonstrate consistent piezoelectric amplitude regardless of polarization direction upon ± 10 V application, supporting the claimed single-domain behavior. However, in Fig. S9, despite nearly identical measurement conditions, the region subjected to -10 V application exhibits a distinctly larger response than the region subjected to $+10$ V. Furthermore, although the application of -10 V from the unpoled state does not induce any change in the polarization direction, an increase in the piezoelectric response is observed, which is in clear contradiction with the single-domain nature claimed on the basis of the TEM observations.

The observed photodetection properties themselves appear genuine and noteworthy. However, the proposed ferroelectric mechanism relies on materials characterization that contains contradictory evidence and lacks internal consistency. Without a clear and consistent experimental foundation, the mechanistic explanation for the exceptional photodetector performance remains insufficiently supported. While the authors have made efforts to address previous concerns, the newly provided data have inadvertently highlighted fundamental issues with experimental reproducibility. In the current form, the manuscript does not meet the standard for Nature Communications.

Version 4:

Reviewer comments:

Reviewer #2

(Remarks to the Author)

The reviewer appreciates the additional PFM data (Fig. S9).

These PFM images demonstrate that the PFM measurements for the pristine regions exhibit a certain degree of reproducibility, which strengthens the basis for discussion.

Based on the revised manuscript, I would like to summarize once again the evidence supporting the ferroelectricity of the SATO thin films:

Observation of the displacement of Al atoms by HAADF

Clear polarization switching behavior in the PUND measurements

Photoresponse characteristics dependent on polarization direction

These results support the conclusion that SATO exhibits ferroelectricity.

On the other hand, the evidence for the single-domain nature of SATO requires more careful evaluation. The authors clearly acknowledge that surface charge injection and changes in surface potential significantly influence the PFM amplitude. Indeed, the results in Fig. S9 show that the SATO film is strongly affected by surface charge injection—an effect much larger than the intrinsic piezoelectric response. Therefore, the possibility of charge-injection-induced contrast cannot be fully excluded, especially because an AC electric field is applied even in the pristine regions during PFM measurements.

Thus, while PFM results may serve as auxiliary information, they should not be treated as conclusive evidence. The description of a single-domain state should be revised to a more modest expression, such as “indicating the possibility of a single domain.”

Furthermore, the presence of surface charge injection also affects the interpretation of the PFM data in Fig. 4. The polarization direction can be meaningfully discussed based on phase only when the piezoelectric response dominates over the surface-charge-related contribution. However, because the SATO film is strongly affected by charge injection, the phase contrast in Fig. 4 should be removed.

I strongly recommend that the primary evidence for ferroelectricity should rely on PUND measurements and structural analysis.

If these revisions are made, the logical consistency of the paper will be greatly improved, and I believe it will reach a level worthy of publication in Nature Communications.

Version 5:

Reviewer comments:

Reviewer #2

(Remarks to the Author)

This paper has undergone multiple rounds of peer review, during which the authors made sincere and appropriate revisions, resulting in a significant improvement in its logical structure. All previously raised concerns have been addressed, and the main conclusions of this study are supported by sufficient experimental evidence.

Now, I'm confident that this paper is sufficiently worthy of publication in Nature Communications. Therefore, I strongly recommend acceptance of this manuscript.

Reply to referees' questions and comments:

Manuscript ID: NCOMMS-25-34130-T

Title: Ferroelectric ultraviolet photodetector material with ultrafast response speed

Authors: Xuexi Yan, Tingting Yan, Lingli Li, Yi Cao, Xinwei Wang, Jinghui Wang, Ang Tao, Tingting Yao, Yixiao Jiang, Weijin Hu, Xiaosheng Fang, Hengqiang Ye, Xiu-Liang Ma, Chunlin Chen

August 2, 2025

First of all, we would like to thank the reviewers for reviewing our manuscript very carefully and putting forth many constructive comments. We have revised our manuscript thoroughly based on these comments. Changes, including minor ones not related to the reviewers' comments, are indicated in red fonts in the revised main text and Supporting Information. Our responses to specific comments follow below:

Reply to referee 1 (R1):

We appreciate the general recognition by the referee that “This manuscript describes the optoelectronic performance of the SATO thin film, which shows a high ultraviolet photodetection sensitivity with a responsivity of 860 mA/W, a detectivity of 1.63×10^{13} Jones, and a switching ratio of 10^4 . The SATO photodetector has an extremely ultrafast response speed with a rise time (τ_r) of 6.8 ns and a fall time (τ_f) of 17.7 ns.”

In the meanwhile, the referee also raises some specific questions and comments which are summarized into six major aspects. We fully understand the referee's concerns, and here we address all the questions and discuss all the comments one-by-one in the following.

Question and comment (R1.1): The manuscript highlights the potential of ferroelectric thin films for high-performance photodetectors, primarily due to their intrinsic electric fields and polarization-related effects. To further support this claim, it would be beneficial to include a detailed analysis of the dielectric properties of the SATO thin film. Parameters such as dielectric constant, dielectric loss, and

their frequency dependence would provide valuable insight into the material's capability for charge storage and screening, which are critical for optimizing photodetector performance. Including these measurements would strengthen the discussion on the multifunctional role of ferroelectricity in optoelectronic applications.

Reply to Question and comment (R1.1):

Thank you very much for your valuable suggestion. According to your suggestions, we performed measurements of the dielectric constant and dielectric loss as a function of frequency in the SATO thin film, as shown in Supplementary Fig. S15. As one can see that the SATO thin film exhibits a large dielectric constant (>16) and low dielectric loss (<0.1) at room temperature. More importantly, the dielectric constant and dielectric loss decrease slowly with increasing frequency, which indicates that the SATO film has good frequency stability. Thus, the large dielectric constant, low dielectric loss, and excellent frequency stability all prove that SATO film has excellent charge storage capacity, which is crucial for its excellent performance in optoelectronic applications.

The following sentences and Supplementary Fig. S15 were added in the revised manuscript and Supplementary Information:

Supplementary Fig. S15 shows that the SATO film has a large dielectric constant (>16), low dielectric loss (<0.1), and excellent frequency stability at room temperature.

Supplementary Fig. S15 | Dielectric constant and dielectric loss as a function of frequency in the SATO thin film. The SATO film exhibits a large dielectric constant, low dielectric loss, and excellent frequency stability at room temperature.

Question and comment (R1.2): The statement "AB₁₂O₁₉ magnetoplumbite compounds have garnered significant attention owing to their multifunctional physical properties, including ferromagnetism, ferroelectricity, high dielectric constant, and wide bandgap. These characteristics make them promising candidates for applications in spintronics, microwave devices, and multifunctional electronic materials." It is superficial in nature, without any strong evidence/reference.

Reply to Question and comment (R1.2):

Thanks for your precious suggestions. We revised the introduction of the manuscript and added application examples of AB₁₂O₁₉ magnetoplumbite to demonstrate its potential as a functional material. The following sentences and references were added in the revised manuscript:

AB₁₂O₁₉ magnetoplumbite compounds have attracted intense interest due to the abundant physical properties. For example, SrFe₁₂O₁₉ is widely used in microwave absorbers, magnetic recording media and sensors, high-frequency electromagnetic (EM) devices, etc. due to its superior magnetic parameters, high magnetic permeability and low electrical conduction loss.²⁰ BaFe₁₂O₁₉ is the best choice for commercial low-cost permanent magnet materials due to its high coercivity and good chemical and thermal stability. It is widely used in motors, speakers, sensors and other devices.²¹ In addition, PbFe₁₂O₁₉ is a natural multiferroic material with broad application prospects in non-volatile memory, ferroelectric photovoltaics and other fields.^{22,23} As a structurally similar derivative, SrAl₁₂O₁₉ was predicted to have ferroelectricity but has not yet been experimentally proved.²⁴

References:

20. Anantharamaiah, P. N., Shashanka, H. M., Saha, S., Haritha, K. & Ramana, C. V. Aluminum Doping and Nanostructuring Enabled Designing of Magnetically Recoverable Hexaferrite Catalysts. *ACS Omega* 7, 6549–6559 (2022).
21. Pullar, R. C. Hexagonal ferrites: A review of the synthesis, properties and applications of hexaferrite ceramics. *Prog. Mater Sci.* 57, 1191–1334 (2012).
22. Tan, G. -L. & Li, W. Ferroelectricity and ferromagnetism of M-type lead hexaferrite. *J. Am. Ceram. Soc.* 98, 1812–1817 (2015).

23. Shen, S.-P. et al. Quantum electric-dipole liquid on a triangular lattice. *Nat. Commun.* **7**, 10569 (2016).
24. Li J., Medina E. A., Stalick J. K., Sleight A. W. & Subramanian M. A. Structural studies of $\text{CaAl}_{12}\text{O}_{19}$, $\text{SrAl}_{12}\text{O}_{19}$, $\text{La}_{2/3+\delta}\text{Al}_{12-\delta}\text{O}_{19}$, and $\text{CaAl}_{10}\text{NiTiO}_{19}$ with the hibonite structure; indications of an unusual type of ferroelectricity. *Z. Naturforsch. B* **71**, 475–484 (2016).

Question and comment (R1.3): The polarization hysteresis data presented in Fig. 4f convincingly demonstrate the ferroelectric nature of the SATO thin film, with a notable remanent polarization (P_r) of $7.8 \mu\text{C}/\text{cm}^2$ and a coercive field (E_c) of 21 MV/m at 300 K and 1 kHz. To further substantiate the intrinsic ferroelectric behavior and rule out potential non-ferroelectric contributions (e.g., from leakage or dielectric effects), it would be valuable to include Positive-Up-Negative-Down (PUND) measurements. Additionally, a fatigue analysis showing the stability of polarization over repeated electric field cycling would strengthen the case for the material's potential in non-volatile memory or other ferroelectric device applications. Including these characterizations would significantly enhance the credibility and completeness of the ferroelectric performance assessment. More importantly, the authors have NOT highlighted the primary cause of the development/origin of polarization; it must be illustrated in the revised manuscript.

Reply to Question and comment (R1.3):

We sincerely appreciate your constructive suggestions on our work. The PUND mode is one of the most accurate methods for measuring the remanent polarization of ferroelectric materials. It can eliminate the influence of dielectric and leakage effects and reflect the intrinsic characteristics of ferroelectricity. Fig. 4f in the manuscript was measured through positive-up-negative-down (PUND) method at 300 K and 1 kHz. Supplementary Fig. S10 shows the measurement parameters of the PUND mode. In the test, the write voltage and read voltage were both 10 V. The delay time between P and U or N and D is fixed 0.5 ms. In addition, we added hysteresis loop measurements at different voltages through the PUND mode, as shown in Supplementary Fig. S11a. It can be found that as the voltage increases, the remanent polarization gradually increases. When the voltage is higher than 10 V, the remanent polarization remains unchanged ($\sim 7.8 \mu\text{C}/\text{cm}^2$), which indicates that the polarization

saturation has been reached.

To explore the potential application of SATO films in non-volatile memory or other ferroelectric micro-devices, ferroelectric fatigue testing was carried out, as shown in Supplementary Fig. S11b. The voltages applied to the test were 10 V and 12 V, respectively. After 10^6 cycles, the remanent polarization intensity did not attenuate significantly, which indicates that the SATO film has a strong fatigue stability.

The origin for the ferroelectricity of SATO has been experimentally identified as the off-centre displacement of Al^{3+} in the AlO_5 bipyramids, as depicted in Supplementary Fig. S12. The polarization of SATO is along the c axis. Thus, the SATO thin film belongs to the first-type displacement-type ferroelectrics.

The following sentences and Supplementary Fig. S10, Fig. S11, and Fig. S12 were added in the revised manuscript and Supplementary Information:

To unequivocally establish the ferroelectric nature of the SATO thin film, macroscopic polarization hysteresis loops were measured through Positive-up-negative-down (PUND) method at 300 K and 1 kHz, as shown in Fig. 4f. The PUND mode is one of the most accurate methods for measuring the remanent polarization of ferroelectric materials. It can eliminate the influence of dielectric and leakage effects and reflect the intrinsic characteristics of ferroelectricity. Supplementary Fig. S10 shows the measurement parameters of the PUND mode. In the test, the write voltage and read voltage were both 10 V.

Hysteresis loops measured under different voltages are shown in Supplementary Fig. S11a. It is clear that the remanent polarization gradually increases with the voltage. After the voltage is increased to 10 V and higher, the remanent polarization remains unchanged ($\sim 7.8 \mu\text{C}/\text{cm}^2$) and the polarization saturation is reached. Supplementary Fig. S11b shows the results of ferroelectric fatigue testing. After 10^6 cycles, the remanent polarization intensity did not attenuate significantly, which indicates that the SATO film has a strong fatigue stability. The atomic origin for the ferroelectricity of SATO has been experimentally identified as the displacement of Al atoms within the AlO_5 bipyramids, as depicted in Supplementary Fig. S12.

Supplementary Fig. S10 | Parameters of positive-up-negative-down (PUND) measurements of the SATO films. The write and read voltages are both 10V.

Supplementary Fig. S11 | Ferroelectric fatigue properties of the SATO thin film tested by PUND measurements. (a) Hysteresis loops of the SATO thin film under different voltages. (b) Fatigue characteristics of the SATO thin film under 10 V and 12 V. Inset are the PUND loops before and after cycling. After 10^6 cycles, the polarization intensity exhibits a strong fatigue stability without significant attenuation.

Supplementary Fig. S12 | Atomic origin of the ferroelectricity of SATO. (a) and (b) HAADF images showing the upward and downward polarities, respectively. The red and green dotted circles represent the O and Al atoms in the AlO_5 bipyramid, respectively. The upward and downward displacement of Al atoms inside the AlO_5 bipyramids leads to the emergence of ferroelectricity.

Question and comment (R1.4): The HAADF and ABF STEM images of the SATO thin film along the $[11\bar{2}0]$, $[1\bar{1}00]$, and $[0001]$ zone axes are clearly presented, and the overlaid atomic models aid significantly in visualizing the local atomic arrangement. Polarization mapping based on the HAADF-STEM data is recommended to enhance the structural analysis further. Quantitative mapping of cation displacements—especially the relative shifts between A-site and B-site cations or between cations and anions—could provide direct insight into the local polarization vectors and their directional variation. This addition would confirm the presence of spontaneous polarization at the atomic scale and establish a stronger structure-property correlation for the observed ferroelectric behavior in SATO.

Reply to Question and comment (R1.4):

Thanks for your valuable suggestion. We used the CalAtom software package (<https://github.com/FangLinF/CalAtom>) and self-developed MATLAB scripts to perform quantitative analysis on the HAADF STEM images, including gaussian filtering of atomic intensity profile, atomic coordinate position identification, element type classification, and calculation of the Al^{3+} offset in the AlO_5 trigonal bipyramid. As shown in supplementary Fig. S13, the direction and length of the arrows

represent the direction of the ferroelectric polarization and displacement, respectively. The length of the arrows has been magnified 25 times to visualize more clearly the displacement of the Al atoms. The displacement of Al atoms in this image ranges from 7 to 15 pm.

The following sentences and Supplementary Fig. S13 were added in the revised Supplementary Information.

As can be seen, the polarization of SATO is vertically upward, indicating that SATO is a uniaxial ferroelectric along the c-axis. Thus, the SATO thin film belongs to the first-type displacement-type ferroelectrics.

Supplementary Fig. S13 | Analysis of the ferroelectric polarization in SATO. (a) HAADF STEM image of SATO along $[11\bar{2}0]$ axis zone. **(b)** Quantitative analysis of polarization displacement in the HAADF image. The direction and length of the green arrows represent the displacement of Al atoms inside the AlO_5 bipyramids. The length of the arrows has been magnified by 25 times to visualize more clearly the displacement of Al atoms. The displacement of Al atoms in this image ranges from 7 to 15 pm. Scale bar, 1nm.

Question and comment (R1.5): In Fig. 5, (a) & (b), legends are not showing clearly, so increase the size; (b) all the graphs overlapped at different power intensities, so enlarge this figure; (c) caption does not mention power intensity; (e) Y-axis label doesn't show clearly, so increase the size, (f) not mentioned wavelength and power intensity, so also mention in the caption.

Reply to Question and comment (R1.5):

Thanks for your suggestion. Fig. 5 has been carefully adjusted and added to the revised manuscript.

Fig. 5 | Photoelectric properties the SATO films. (a) Logarithmic I-V curves of the SATO PDs under different light wavelengths (250-470 nm). The SATO PDs have the strongest response to 330 nm UV light. (b) I-V curves of the SATO PDs under different power density at 330 nm. (c) I-T curves under 330 nm on-off illumination at -2, -6, and -10 V of the SATO PDs. (d) Responsivity, detectivity, and (e) On/off ratio curves at different wavelengths. At 330 nm, the responsivity, detectivity, and on/off ratio of the SATO PDs at -2 V are as high as 860 mA/W, 1.63×10^{13} Jones, and 1.9×10^4 , respectively. (f) Time-resolved transient photoresponse curves. The SATO PDs exhibits extremely fast response speed with rise time (τ_r) of 6.8 ns and fall time (τ_f) of 17.7 ns.

Question and comment (R1.6): The transient photoconductivity measurements and their results make this paper unique and might deserve publication in Nature Communications. However, the authors

have just put one graph, Fig. 5 (f), for the same. It requires its repeatability (long time ON/OFF), current intensity (absolute value), different laser intensity, device effective area (at least three sizes of the device), etc., to calculate the responsivity and detectivity of the photodetector (PD) as claimed!! I would be happy to see the detailed analysis of Fig. 5 (f), which supports and corroborates the claims. A table (supplementary) is needed for all the inputs with responsivity and detectivity calculations.

Reply to Question and comment (R1.5):

Thank you very much for your recognition and high evaluation on this study and thanks for your precious suggestion. First, we added the long-term repeated on-off switching measurements of the SATO photodetector, as shown in Supplementary Fig. S16. The measurement was carried out under 330 nm laser illumination with a power density of $50 \mu\text{W}/\text{cm}^2$. As one can see, the currents of the SATO photodetector remained almost unchanged during the test process of up to 2000s, which indicates that the SATO photodetector has an excellent reproducibility and detection reliability.

Photodetectors with groove widths ranging from $5 \mu\text{m}$ to $17 \mu\text{m}$ were fabricated by a direct-write lithography system, as shown in Supplementary Fig. S17. To evaluate the performance of the photodetector, we selected three devices with groove widths of $5 \mu\text{m}$, $8 \mu\text{m}$, and $11 \mu\text{m}$ for systematic measurement. That is, the corresponding effective area of these devices are $2.5 \times 10^{-6} \text{ cm}^2$ (Device 1), $4.0 \times 10^{-6} \text{ cm}^2$ (Device 2), and $5.5 \times 10^{-6} \text{ cm}^2$ (Device 3). Then, we measured the three devices at different laser power densities and calculated the Responsivity (R) and detectivity (D^*), as shown in Supplementary Fig. S18. It can be found that the devices with three effective areas all have excellent weak light detection capabilities as low as $50 \mu\text{W}/\text{cm}^2$.

In order to determine the effect of polarization direction on the photoelectric response of SATO thin films, we pre-polarized the AlN thin films using high voltage before testing. As shown in the Supplementary Fig. S19, the currents formed by the SATO photodetectors pre-polarized at 30V and -30V under 330 nm illumination are almost equal, which indicates that the effects of positive and negative polarization on the performance of the photodetector are actually equivalent. In addition, we have added time-resolved transient photoresponse measurements dependent on the polarization direction, as shown in Supplementary Fig. S20. As one can see that the SATO photodetector has a stable photoresponse regardless of whether it is in the positive or negative polarization state, as shown

in Supplementary Fig. S20a and d. More importantly, the SATO photodetectors with positive and negative polarizations have ultra-fast response speed with rise and fall times of 6.6 ns/17.7 ns and 8.7ns/12.2 ns, respectively (Supplementary Figs. S20), which is basically consistent with the response speed of the unprepolarized SATO photodetector (Fig. 5f in the main text).

Finally, we made statistics on the current intensity, effective area, laser intensity, responsivity, detectivity and response time of SATO photodetector devices, as shown in the Supplementary Table S1. All the SATO photodetector devices we measured have extremely fast response speed, which can greatly reduce the response delay of functional devices and improve the efficiency of information exchange and detection.

The following sentences and Supplementary Fig. S16 to Fig. S20 were added in the revised manuscript and Supplementary Information:

Furthermore, long-term repeated on-off switching measurements shown in supplementary Fig. S16 reveal that the photocurrent and dark current remain stable, and the SATO detector has the excellent reproducibility and detection reliability.

To quantitatively evaluate the performance of the SATO PDs, we prepared three devices with different effective areas (Supplementary Fig. S17)

Supplementary Fig. S18 shows the responsivity and detectivity of different devices at different laser power intensity under 330 nm illumination. The three devices exhibit the same feature. The responsivity and detectivity gradually decrease with the increase of laser intensity.

The photoelectric switching response and transient photoelectric response of SATO with different polarization directions were measured. As shown in Supplementary Fig. S19, the SATO photodetector has a strong and stable photoelectric response regardless of positive or negative polarization. More importantly, as shown in Supplementary Figs. S20, the SATO photodetectors with positive and negative polarizations have ultra-fast response speed. Their rise/fall times are 6.6 ns/17.7 ns and 8.7ns/12.2 ns, respectively. Similar tests using other devices with different effective area were carried out. The current intensity, effective area, laser intensity, responsivity, detectivity and response time of three SATO photodetector devices are listed the Supplementary Table S1. All the SATO photodetector devices exhibit extremely fast response speed.

Supplementary Fig. S16 | Long-term stability of the SATO photodetector under 330 nm laser illumination with a power intensity of $50 \mu\text{W}/\text{cm}^2$. There is almost no significant change in the current, indicating the excellent stability and reliability of the SATO photodetector.

Supplementary Fig. S17 | Optical microscope images of the SATO photodetectors fabricated by photolithography. The size of the Pt electrode is $50 \mu\text{m} \times 50 \mu\text{m}$. Scale bar, $500 \mu\text{m}$ in (a) and $50 \mu\text{m}$ in (b).

Supplementary Fig. S18 | Responsivity and detectivity curves at different laser power density.

All the devices have excellent detection capabilities of weak light.

Supplementary Fig. S19 | I-T curves under 330 nm on-off illumination at positive and negative polarizations of the SATO photodetector. The photocurrents under positive and negative polarizations are almost equal.

Supplementary Fig. S20 | Time-resolved transient photoresponse curves of the SATO photodetectors with positive and negative polarizations. The SATO photodetectors with positive and negative polarizations have ultra-fast response speed. Their rise/fall times are 6.6 ns/17.7 ns (a-c) and 8.7ns/12.2 ns (d-e), respectively.

Table S1 | Statistical summary of the optoelectronic performance of SATO photodetectors.

	I_d (nA)	I_p (nA)	S (cm^2)	P ($\mu\text{W}/\text{cm}^2$)	R (mA/W)	D^* (Jones)	Response time (ns)
Device 1 (No polarization)	2.10×10^{-5}	1.08×10^{-1}	2.5×10^{-6}	50	860.00	1.63×10^{13}	6.8/17.7
Device 1 (Positive polarization)	2.25×10^{-5}	1.09×10^{-1}	2.5×10^{-6}	50	871.82	1.62×10^{13}	6.6/14.7
Device 1 (Negative polarization)	1.80×10^{-5}	9.08×10^{-2}	2.5×10^{-6}	50	726.26	1.51×10^{13}	8.7/12.2
Device 2	4.02×10^{-5}	1.53×10^{-1}	4.0×10^{-6}	50	764.67	1.35×10^{13}	9.9/21.3
Device 3	5.15×10^{-5}	1.75×10^{-1}	5.5×10^{-6}	50	636.18	1.16×10^{13}	15.5/30.0

Sincerely thank you very much for your constructive comments, suggestions and valuable questions. Your comments and important questions were very helpful in improving our manuscript.

Reply to referee 2 (R2):

We appreciate the general recognition by the referee that “This manuscript presented SrAl₁₁TiO₁₉ (SATO) thin films as a novel ferroelectric material for ultrafast ultraviolet photodetectors. While the TEM structural analysis is generally well-executed, the ferroelectric characterization requires substantial reinforcement.” In the meanwhile, the referee also raises some specific questions and comments. We fully understand the referee’s concerns, and address all the comments one-by-one in the following.

Major Concerns

Question and comment (R2.1): Inadequate Ferroelectric Characterization:

P-E Hysteresis Analysis: The P-E loops presented in Figure 4f lack essential electric field dependence data. Demonstrating polarization saturation at high fields is critical for confirming intrinsic ferroelectric switching behavior. The current data are insufficient to distinguish between true ferroelectric switching and other polarization mechanisms.

PFM Domain Analysis: While the PFM images successfully demonstrate voltage-induced domain writing (Figures 4b and 4d), the study lacks PFM characterization of pristine, as-grown film regions. This baseline comparison is crucial for determining whether the observed ferroelectricity represents an intrinsic material property or results from field-induced effects.

Reply to Question and comment (R2.1):

Thank you very much for your precious suggestion. To reveal the intrinsic ferroelectricity of SATO thin films, we performed P-E loops measurements at different voltages through the Positive-up-negative-down (PUND) mode, as shown in Supplementary Fig. S11a. It can be found that as the voltage increases, the remanent polarization gradually increases. When the voltage is greater than 10 V, the remanent polarization is stable ($\sim 7.8 \mu\text{C}/\text{cm}^2$), which indicates that polarization saturation has been reached. In addition, to reveal that SATO has stable intrinsic ferroelectricity, we carried out ferroelectric fatigue measurements, as shown in Supplementary Fig. S11b. The voltages applied to the test were 10 V and 12 V, respectively. After 10^6 cycles, the remanent polarization intensity did not attenuate significantly, which indicates that the SATO film has a strong fatigue stability.

To exclude the possibility that the ferroelectricity comes from the pristine, as-grown AlN thin film on the Nb-doped STO substrate, we performed PFM characterizations, as shown in Supplementary Fig. S9. Electrode domains cannot be written in the AlN film, which demonstrates that AlN is not a ferroelectric material. On the other hand, low-magnification TEM bright-field images and corresponding selected area electron diffraction (SAED) characterization indicate that the AlN film has completely phase-changed with the STO substrate and transformed into a SATO single-phase film after high-temperature annealing in air, as shown in Fig. 1 in the main text and Supplementary Fig. S6 in Supplementary Information. Both performance testing and structural characterization can prove that the ferroelectricity comes from the SATO film, not the initially grown AlN film.

The following sentences and Supplementary Fig. S11 and Fig. S9 were added in the revised manuscript and Supplementary Information.

Hysteresis loops measured under different voltages are shown in Supplementary Fig. S11a. It is clear that the remanent polarization gradually increases with the voltage. After the voltage is increased to 10 V and higher, the remanent polarization remains unchanged ($\sim 7.8 \mu\text{C}/\text{cm}^2$) and the polarization saturation is reached. Supplementary Fig. S11b shows the results of ferroelectric fatigue testing. After 10^6 cycles, the remanent polarization intensity did not attenuate significantly, which indicates that the SATO film has a strong fatigue stability.

To exclude the possibility that the ferroelectricity comes from the AlN thin film grown on the Nb:STO substrate, we performed PFM characterizations and showed the results in Supplementary Fig. S9. As can be seen, it is difficult to write domains in the AlN film, which demonstrates that the AlN thin film is not ferroelectric.

Supplementary Fig. S11 | Ferroelectric fatigue properties of the SATO thin film tested by PUND measurements. (a) Hysteresis loops of the SATO thin film under different voltages. (b) Fatigue characteristics of the SATO thin film under 10 V and 12 V. Inset are the PUND loops before and after cycling. After 10^6 cycles, the polarization intensity exhibits a strong fatigue stability without significant attenuation.

Supplementary Fig. S9 | PFM characterizations of the AlN thin film on the Nb:STO (111) substrate. (a-c) PFM topography, amplitude, and phase images plots of the AlN film. Domains cannot be written in the AlN film, suggesting that the AlN film is not ferroelectric. Scale bar, 2 μm .

Question and comment (R2.2): Stoichiometry and Charge Balance Issues:

The reported SATO stoichiometry raises fundamental questions about charge neutrality. The manuscript must address how electronic balance is maintained in this composition—specifically, the oxidation state of titanium and the potential role of intrinsic defects. This analysis is crucial for understanding both the insulating properties and ferroelectric behavior of SATO.

Reply to Question and comment (R2.2):

Thanks for your valuable this comment. Your suggestions have led us to think deeply on the charge balance issues in SATO. SATO can be understood as Ti atoms partially replacing the Al atoms in $\text{SrAl}_{12}\text{O}_{19}$ magnetoplumbite. Therefore, determining the valence state of Ti is key to understanding the stoichiometry and charge balance of SATO. Here, we characterized the Ti $L_{2,3}$ fine structure using atomic-resolved electron energy-loss spectroscopy (EELS) with an energy resolution of 0.3 eV, as shown in Supplementary Fig. S5. It can be clearly found that the EELS spectra of SATO and STO have the same fine structures with four peaks, indicating that the valence of Ti in SATO is +4. To maintain

the charge neutrality of the SATO structure, Al vacancies must exist in SATO. Thank you again for your insightful suggestions.

The sentences and Supplementary Fig. S5 were added in the revised manuscript and Supplementary Information.

This compound is derived from $\text{SrAl}_{12}\text{O}_{19}$ after replacing one Al atom by Ti. To understand the stoichiometry and charge balance of SATO, electron energy-loss spectroscopy (EELS) analysis was carried out. Ti $L_{2,3}$ edges of SATO and STO are shown in Supplementary Fig. S5. It is clear that the Ti $L_{2,3}$ edges of SATO and STO have very similar fine structures with four peaks, suggesting that the Ti ions in SATO have the valence state of +4. To maintain the charge neutrality of the SATO structure, Al vacancies must exist in SATO.

Supplementary Fig. S5 | EELS spectra showing the Ti $L_{2,3}$ edges in STO and SATO. The Ti $L_{2,3}$ edges of SATO and STO have very similar fine structures with four peaks, suggesting that the Ti ions in SATO have the valence state of +4.

Question and comment (R2.3): Substrate Interaction Effects:

The SATO films are grown on Nb-doped STO substrates and processed at 1500C. At these elevated temperatures, Nb diffusion from the substrate into the SATO layer is highly probable. The authors must evaluate this possibility and discuss its potential impact on film conductivity, ferroelectric properties, and photoresponse characteristics.

Reply to Question and comment (R2.3):

Sincerely thanks for your valuable comment. To determine whether there is diffusion of Nb from the substrate to the SATO thin film or not, we used the EDS spectra obtained in the aberration-corrected STEM to characterize the elements and content in the SATO film. Supplementary Fig. S4 shows the EDS spectra of the 0.7at.% Nb: STO substrate and the SATO thin film. It can be found that except for the peaks of Sr, Ti, and O, there is also an obvious peak of Nb in the EDS spectrum of the substrate, as shown by the arrows in Supplementary Fig. S4a. In the SATO film, only the peaks of Sr, Al, Ti, and O are observed and no Nb peak can be detected, as shown in Supplementary Fig. S4b. This indicates that the Nb element in the substrate does not diffuse into the film during sample preparation.

Supplementary Fig. S4 | EDS spectrum recorded from the 0.7% Nb: STO (111) substrate and SATO thin film. There are only Sr, Al, Ti, and O elements in the SATO thin film and no Nb element diffused from the substrate.

Question and comment (R2.4): Polarization-Dependent Photoresponse:

A defining characteristic of ferroelectric photodetectors is their ability to exhibit polarization-switchable performance. The study lacks critical experiments demonstrating how photoresponse parameters (photocurrent magnitude, responsivity, response time) vary with deliberate polarization reversal in the SATO film. Such measurements would provide compelling evidence for ferroelectricity-enhanced photodetection.

Reply to Question and comment (R2.4):

We sincerely appreciate your constructive suggestions on our work. We prepolarized the SATO film using high voltage in different directions and measured the effect of different polarization directions on the photoresponse. As shown in the Supplementary Fig. S19, the currents formed by the SATO photodetectors pre-polarized at 30V and -30V under 330 nm illumination are almost the same, which indicates that the effects of positive and negative polarization on the performance of the photodetector are actually equivalent. In addition, we have added time-resolved transient photoresponse measurements dependent on the polarization direction, as shown in Supplementary Fig. S20. As one can see that the SATO photodetector has a stable photoresponse regardless of whether it is in the positive or negative polarization state, as shown in Supplementary Fig. S20a and d. More importantly, the SATO photodetectors with positive and negative polarizations have ultra-fast response speed with rise and fall times of 6.6 ns/17.7 ns and 8.7ns/12.2 ns, respectively (Supplementary Figs. S20b, c, e, and f), which is basically consistent with the response speed of the no prepolarized SATO photodetector (Fig. 5f in the main text).

Similarly, we performed the same test on other devices with different effective area and statistically summarized on the current intensity, effective area, laser intensity, responsivity, detectivity and response time of SATO photodetector devices, as shown in the Supplementary Table S1. All the SATO photodetector devices we measured have extremely fast response speed, which can greatly reduce the response delay of functional devices and improve the efficiency of information exchange and detection.

The following sentences and Supplementary Fig. S19, Fig. S20, and Table S1 were added in the revised manuscript and Supplementary Information.

The photoelectric switching response and transient photoelectric response of SATO with different polarization directions were measured. As shown in Supplementary Fig. S19, the SATO photodetector has a strong and stable photoelectric response regardless of positive or negative polarization. More importantly, as shown in Supplementary Figs. S20, the SATO photodetectors with positive and negative polarizations have ultra-fast response speed. Their rise/fall times are 6.6 ns/17.7 ns and 8.7ns/12.2 ns, respectively. Similar tests using other devices with different effective area were carried out. The current intensity, effective area, laser intensity, responsivity, detectivity and response time of three SATO photodetector devices are listed the Supplementary Table S1. All the SATO photodetector devices exhibit extremely fast response speed.

Supplementary Fig. S19 | I-T curves under 330 nm on-off illumination at positive and negative polarizations of the SATO photodetector. The photocurrents under positive and negative polarizations are almost equal.

Supplementary Fig. S20 | Time-resolved transient photoresponse curves of the SATO photodetectors with positive and negative polarizations. The SATO photodetectors with positive and negative polarizations have ultra-fast response speed. Their rise/fall times are 6.6 ns/17.7 ns (a-c) and 8.7ns/12.2 ns (d-e), respectively.

Table S1 | Statistical summary of the optoelectronic performance of SATO photodetectors.

	I_d (nA)	I_p (nA)	S (cm^2)	P ($\mu\text{W}/\text{cm}^2$)	R (mA/W)	D^* (Jones)	Response time (ns)
Device 1 (No polarization)	2.10×10^{-5}	1.08×10^{-1}	2.5×10^{-6}	50	860.00	1.63×10^{13}	6.8/17.7
Device 1 (Positive polarization)	2.25×10^{-5}	1.09×10^{-1}	2.5×10^{-6}	50	871.82	1.62×10^{13}	6.6/14.7
Device 1 (Negative polarization)	1.80×10^{-5}	9.08×10^{-2}	2.5×10^{-6}	50	726.26	1.51×10^{13}	8.7/12.2
Device 2	4.02×10^{-5}	1.53×10^{-1}	4.0×10^{-6}	50	764.67	1.35×10^{13}	9.9/21.3
Device 3	5.15×10^{-5}	1.75×10^{-1}	5.5×10^{-6}	50	636.18	1.16×10^{13}	15.5/30.0

Minor Concerns

Question and comment (R2.5): Device Visualization: Including a photograph of the fabricated SATO photodetector would help readers better understand the device architecture and experimental configuration.

Reply to Question and comment (R2.5):

Sincerely thanks for your suggestion. We used a direct-write lithography system to fabricate photodetectors with groove widths ranging from 5 μm to 17 μm , as shown in Supplementary Fig. S17. In this work, we selected photodetectors with groove widths of 5 μm , 8 μm , and 11 μm to evaluate their performance.

The following Supplementary Fig. S17 was added in the revised Supplementary Information.

Supplementary Fig. S17 | Optical microscope images of the SATO photodetectors fabricated by photolithography. The size of the Pt electrode is $50\ \mu\text{m} \times 50\ \mu\text{m}$. Scale bar, $500\ \mu\text{m}$ in (a) and $50\ \mu\text{m}$ in (b).

Question and comment (R2.6): Domain Structure Representation: The schematic diagram of ferroelectric domains in Supplementary Fig. S1 does not accurately reflect typical domain patterns observed in ferroelectric thin films. Domain illustrations should be grounded in experimental observations from either the literature or the authors' work to ensure scientific accuracy and educational value.

Reply to Question and comment (R2.6):

Sincerely thanks you very much for this comment. We have redrawn the schematic diagram of the ferroelectric domain structure based on the experimental results, as shown in Supplementary Fig. S1.

Supplementary Fig.S1 was revised in the revised Supplementary Information.

Supplementary Fig. S1 | Schematic diagram of the performance failure of traditional ferroelectric photodetectors. (a) Spontaneous polarization and (b) built-in electric field distribution in traditional ferroelectric materials. Most of ferroelectric materials have a high density of ferroelectric domains with different polarization directions. (c-f) Several common failure mechanisms of ferroelectric photodetectors include scattering, annihilation, capture of photogenerated carriers at domain walls, and cancellation of built-in electric fields between different domains.

Reference

1. Yang, S. Y. *et al.* Above-bandgap voltages from ferroelectric photovoltaic devices. *Nat. Nanotechnol.* **5**, 143–147 (2010).

Question and comment (R2.7): Addressing these concerns—particularly the ferroelectric characterization gaps and stoichiometry questions—will substantially strengthen the manuscript's scientific rigor and enhance the credibility of claims regarding this novel ferroelectric photodetector material.

Reply to Question and comment (R2.7):

Thank you very much for your valuable comments of our work. We fully understand the referee's concerns and address all the questions one by one based on your comments.

We sincerely thank the reviewers for their constructive comments, suggestions, and valuable questions. Your comments and important questions were very helpful in improving our manuscript.

Reply to referee 3 (R3):

We appreciate the general recognition by the referee that “Yan et al present an outstanding optoelectronic performance of the single-domain ferroelectric SATO thin film. The ferroelectric properties have been discussed and they provide excellent optoelectronic performances for ultraviolet region, such as responsibility of 860mA/W, the rise time of 6.8 ns and the fall time of 17.7 ns. I think the paper is well organized, and the presented experimental results are very interesting.”

In the meanwhile, the referee also raises some specific questions and comments which are summarized into four major aspects. We fully understand the referee’s concerns, and here we address all the questions and discuss all the comments one-by-one in the following.

Question and comment (R3.1): The authors used the PFM and polarization hysteresis loops to show the ferroelectric behavior of SATO thin film. While the authors state that the high dielectric constants of ferroelectric materials lead to lower noise level, it will be helpful if they can show dielectric behavior of SATO thin film and compare with other ferroelectric materials.

Reply to Question and comment (R3.1):

Thank you very much for your valuable suggestion. According to your suggestions, we performed measurements of the dielectric constant and dielectric loss as a function of frequency in the SATO thin film, as shown in Supplementary Fig. S15. The SATO thin film exhibits a high dielectric constant (>16) and low dielectric (<0.1) at room temperature, which indicates that the SATO film is a material with strong polarity. More importantly, the dielectric constant and dielectric loss decrease slowly with increasing frequency, which indicates that the SATO film has a good frequency stability. Thus, the large dielectric constant, low dielectric loss, and excellent frequency stability all prove that SATO film has excellent charge storage capacity, which is crucial for its excellent performance in optoelectronic applications.

The following sentences and Supplementary Fig. S15 were added in the revised manuscript and Supplementary Information.

Supplementary Fig. S15 shows that the SATO film has a large dielectric constant (>16), low dielectric loss (<0.1), and excellent frequency stability at room temperature.

Supplementary Fig. S15 | Dielectric constant and dielectric loss as a function of frequency in the SATO thin film. The SATO film exhibits a large dielectric constant, low dielectric loss, and excellent frequency stability at room temperature.

Question and comment (R3.2): The authors state that the ferroelectric polarization induces intrinsic built-in electric field increasing optoelectronic properties. I am wondering whether the ferroelectricity can be tailored in SATO thin film, that the effect of ferroelectric polarization can be confirmed. They could measure optoelectronic performances using the SATO thin films with different spontaneous polarization.

Reply to Question and comment (R3.2):

We sincerely appreciate your constructive suggestions on our work. We prepolarized the SATO film using high voltage in different directions and measured the effect of different polarization directions on the photoresponse. As shown in the Supplementary Fig. S19, the currents formed by the SATO photodetectors pre-polarized at 30V and -30V under 330 nm illumination are almost the same, which indicates that the effects of positive and negative polarization on the performance of the photodetector are actually equivalent. In addition, we have added time-resolved transient photoresponse measurements dependent on the polarization direction, as shown in Supplementary Fig. S20. As one can see that the SATO photodetector has a stable photoresponse regardless of whether it is in the positive or negative polarization state, as shown in Supplementary Fig. S20a and d. More importantly,

the SATO photodetectors with positive and negative polarizations have ultra-fast response speed with rise and fall times of 6.6 ns/17.7 ns and 8.7ns/12.2 ns, respectively (Supplementary Figs. S20b, c, e, and f), which is basically consistent with the response speed of the no prepolarized SATO photodetector (Fig. 5f in the main text).

Similarly, we performed the same test on other devices with different effective area and statistically summarized on the current intensity, effective area, laser intensity, responsivity, detectivity and response time of SATO photodetector devices. The results are shown in the Supplementary Table S1. All the SATO photodetector devices we measured have extremely fast response speed, which can greatly reduce the response delay of functional devices and greatly improve the efficiency of information exchange.

The following sentences and Supplementary Fig. S19, Fig. S20, and Table S1 were added in the revised manuscript and Supplementary Information.

The photoelectric switching response and transient photoelectric response of SATO with different polarization directions were measured. As shown in Supplementary Fig. S19, the SATO photodetector has a strong and stable photoelectric response regardless of positive or negative polarization. More importantly, as shown in Supplementary Figs. S20, the SATO photodetectors with positive and negative polarizations have ultra-fast response speed. Their rise/fall times are 6.6 ns/17.7 ns and 8.7ns/12.2 ns, respectively. Similar tests using other devices with different effective area were carried out. The current intensity, effective area, laser intensity, responsivity, detectivity and response time of three SATO photodetector devices are listed the Supplementary Table S1. All the SATO photodetector devices exhibit extremely fast response speed.

Supplementary Fig. S19 | I-T curves under 330 nm on-off illumination at positive and negative polarizations of the SATO photodetector. The photocurrents under positive and negative polarizations are almost equal.

Supplementary Fig. S20 | Time-resolved transient photoresponse curves of the SATO photodetectors with positive and negative polarizations. The SATO photodetectors with positive and negative polarizations have ultra-fast response speed. Their rise/fall times are 6.6 ns/17.7 ns (a-c) and 8.7ns/12.2 ns (d-e), respectively.

Table S1 | Statistical summary of the optoelectronic performance of SATO photodetectors.

	I_d (nA)	I_p (nA)	S (cm^2)	P ($\mu W/cm^2$)	R (mA/W)	D^* (Jones)	Response time (ns)
Device 1 (No polarization)	2.10×10^{-5}	1.08×10^{-1}	2.5×10^{-6}	50	860.00	1.63×10^{13}	6.8/17.7
Device 1 (Positive polarization)	2.25×10^{-5}	1.09×10^{-1}	2.5×10^{-6}	50	871.82	1.62×10^{13}	6.6/14.7
Device 1 (Negative polarization)	1.80×10^{-5}	9.08×10^{-2}	2.5×10^{-6}	50	726.26	1.51×10^{13}	8.7/12.2
Device 2	4.02×10^{-5}	1.53×10^{-1}	4.0×10^{-6}	50	764.67	1.35×10^{13}	9.9/21.3
Device 3	5.15×10^{-5}	1.75×10^{-1}	5.5×10^{-6}	50	636.18	1.16×10^{13}	15.5/30.0

Question and comment (R3.3): It was demonstrated that the single-domain ferroelectric thin films can result the high optoelectronic performances. Is all single-domain ferroelectric thin films are possible to result the high optoelectronic performances? If not, what distinguishes SATO from other candidates? A brief comparison with other single-domain ferroelectric materials would help contextualize the advantages of the SATO system.

Reply to Question and comment (R3.3):

We appreciate the valuable comments. Generally speaking, we believe that single-domain ferroelectric films can improve the performance of photodetectors, especially the response speed. On one hand, lattice defects such as domain boundaries often generate defect levels, which act as traps for photogenerated carriers, shortening their lifetime and reducing photoelectric conversion efficiency. On the other hand, defects such as domain boundaries can sometimes cause the material to be conducting and increase leakage current, which increases the dark current of the detector and affects its detection accuracy and signal-to-noise ratio. Furthermore, defects such as domain boundaries can scatter photogenerated carriers, hindering their efficient transport and reducing carrier mobility, leading to a decrease in optoelectronic performance. However, not all single-domain ferroelectric thin films necessarily exhibit a high optoelectronic performance. This is because optoelectronic performance is influenced by multiple factors, including the material's band structure, bandgap, and defect density. Even with a single-domain structure, achieving high optoelectronic performance is difficult if the

material is unfavorable for light absorption.

We believe that the main reasons for SATO's excellent performance are as follows: First, its ferroelectric polarization is theoretically only along the c-axis, which makes the migration path of photogenerated carriers unidirectional, resulting in higher photoelectric conversion efficiency and faster response speed; on the other hand, SATO has a large band gap and only absorbs ultraviolet light, so the environmental noise is low and the signal-to-noise ratio is stronger; in addition, through a large number of SATO growth experiments, we found that there are almost no other defects such as dislocations, twins, stacking faults, etc. in its lattice, which shows that SATO is easy to obtain high-quality single crystals and the photoelectric performance is closer to the theoretical value.

Question and comment (R3.4): The SATO-based device demonstrates a nanosecond-scale response time, which is remarkable. What could be origin of the ultrafast response speed compared to other conventional photodetectors?

Reply to Question and comment (R3.4):

Thank you very much for your recognition and high evaluation on this study and thanks for your precious suggestion. The ultrafast response speed of SATO, which is nearly 10,000 times faster than conventional ferroelectric photodetectors (e.g., BaTiO₃, PbTiO₃), originates from the unique structural and physical properties of the SATO thin film.

First, SATO only has ferroelectric polarization along the c direction. This structure can greatly shorten the migration path of photogenerated carriers, which is crucial to improving the response speed. Conventional ferroelectric materials (*ie.*, BaTiO₃, PbTiO₃) typically exhibit high-density ferroelectric domains with random polarization directions. Domain walls in these materials act as scattering centers for photogenerated carriers, which will prolong carrier transport paths, cause carrier annihilation, and significantly slow down the response speed.

Second, SATO has a strong and stable built-in electric field, as shown in and Fig. 4e in main text and Fig. S11 in the Supplementary Information, which facilitates the efficient separation of electron-hole pairs and minimizes the time required for photocurrent generation and decay. In multi-domain ferroelectrics, the built-in electric fields from domains with opposite polarizations partially cancel each other, weakening the net field and reducing the separation efficiency of carriers.

More importantly, SATO can be easily grown into high-quality epitaxial films almost without boundaries and dislocations, as shown in Fig. 1c and Supplementary Fig. S7. Such high-quality growth reduces defect-induced carrier trapping, which is a major cause of slow response in conventional photodetectors. Thus, carriers in SATO can transport without significant trapping, which will accelerate the response.

The following sentences and Supplementary Fig. S11 were added in the revised manuscript and Supplementary Information.

Hysteresis loops measured under different voltages are shown in Supplementary Fig. S11a. It is clear that the remanent polarization gradually increases with the voltage. After the voltage is increased to 10 V and higher, the remanent polarization remains unchanged ($\sim 7.8 \mu\text{C}/\text{cm}^2$) and the polarization saturation is reached. Supplementary Fig. S11b shows the results of ferroelectric fatigue testing. After 10^6 cycles, the remanent polarization intensity did not attenuate significantly, which indicates that the SATO film has a strong fatigue stability.

Supplementary Fig. S11 | Ferroelectric fatigue properties of the SATO thin film tested by PUND measurements. (a) Hysteresis loops of the SATO thin film under different voltages. (b) Fatigue characteristics of the SATO thin film under 10 V and 12 V. Inset are the PUND loops before and after cycling. After 10^6 cycles, the polarization intensity exhibits a strong fatigue stability without significant attenuation.

We sincerely thank all the reviewers for the precious comments, suggestions and questions. Your comments and important questions really help us improve this manuscript, and encourage us

continuing an in-depth and all-round research in this subject. We do hope that the detailed response given to the points made above goes some way towards addressing the comments and the revised manuscript can fit your requirements.

Reply to referees' questions and comments:

Manuscript ID: NCOMMS-25-34130A

Title: Ferroelectric ultraviolet photodetector material with ultrafast response speed

Authors: Xuexi Yan, Tingting Yan, Lingli Li, Yi Cao, Xinwei Wang, Jinghui Wang, Ang Tao, Tingting Yao, Yixiao Jiang, Weijin Hu, Xiaosheng Fang, Hengqiang Ye, Xiu-Liang Ma, Chunlin Chen

September 2, 2025

We would like to thank all reviewers for reviewing our manuscript very carefully and acknowledging our work. We have further revised our manuscript based on the comments and suggestions from Reviewers, which is indicated in red font in the revised main text and Supplementary information.

Reply to referee 1 (R1):

Question and comment (R1.1): The authors have improved the manuscript in line with queries, comments, and concerns raised by reviewers; however, I am still not convinced why these SATO thin films show an increase in coercive field even after reaching of maximum polarization. An ideal ferroelectric system shows nearly the same coercive field after polarization saturation. Polarization graphs also show nearly square hysteresis without saturation (as can be seen PFM data saturation after 12 V); it requires some explanation by the authors.

Reply to Question and comment (R1.1):

We sincerely appreciate your recognition of our work and your precious comments. For ideal ferroelectric materials, the homogeneity of their microstructures, absence of defects, and uniformity in ferroelectric domain switching barriers enable them to maintain a constant coercivity after their hysteresis loops reach saturation. However, defects in ferroelectric materials will induce a notable lateral broadening of the hysteresis loop as the applied electric field or frequency increases beyond the saturation threshold, which in turn leads to a gradual increase in coercivity.^{1,2,3} This phenomenon has been observed in many practical systems and the related physical mechanism has been discussed and

clarified.

The factors contributing to the increase in coercivity of ferroelectric materials include defect pinning, non-uniform electric field distribution, and leakage current coupled with charge injection, among which defect pinning is the most prevalent mechanism. Specifically, intrinsic defects (e.g., vacancies) and extrinsic defects (e.g., heterointerfaces) within the material can act as "pinning centers" that anchor ferroelectric domain walls, thereby impeding their dynamic switching behavior.^{4,5,6} At relatively low applied voltages, ferroelectric domains only need to overcome the constraint of weak pinning sites to complete switching. However, as the voltage increases to higher levels, additional energy input is required to surmount the stronger pinning effects imposed by strong pinning defects (e.g., deep-trap defects), thereby manifesting as an increase in coercivity.

As illustrated in our experimental results (Supplementary Fig. S12), when the applied voltage was increased from 10 V to 12 V, the P-E loop exhibited a significant lateral broadening, while the saturation polarization remained nearly unchanged. This experimental observation is well-aligned with the characteristic behaviors of the defect pinning mechanism.

Al vacancies must exist in SATO films to achieve charge balance. SATO is derived from $\text{SrAl}_{12}\text{O}_{19}$ due to partial substitution of Al atoms by Ti atoms. Therefore, determining the valence state of Ti is key to understanding the stoichiometry and charge balance of SATO. We characterized the Ti $L_{2,3}$ fine structure using atomic-resolved electron energy-loss spectroscopy (EELS) with an energy resolution of 0.3 eV, as shown in Supplementary Fig. S4. It is clear that the EELS spectra of SATO and STO exhibit the same fine structures with four peaks, indicating that the valence of Ti in SATO is +4. To maintain the charge neutrality, Al vacancies must exist in SATO. We define the chemical formula of SATO as $\text{SrAl}_{11-\delta}\text{TiO}_{19}$, where δ represents a parameter related to Al vacancies. To more accurately determine the chemical composition in the SATO thin film, we measured the compositions of several different regions in the SATO film, as shown in Supplementary Table S1. After background subtraction, normalization, and integration of the EDS spectra, the average atomic ratios of Sr, Al, Ti and O in the SATO film is 1.00:10.71:1.02:19.02. The atomic ratio of Sr to Ti is close to 1, similar as that in the SrTiO_3 substrate. Therefore, the δ associated with the Al vacancies in the present SATO film is approximately 0.3, which also satisfies the principle of electrical neutrality. We believe that the Al

vacancies in the SATO film may be the reason for the lateral broadening of the hysteresis loop as the electric field continues to increase after reaching saturation.

The following sentences and Supplementary Fig. S9 were added in the revised Supplementary Information:

Defects in ferroelectric materials will induce a notable lateral broadening of the hysteresis loop as the applied electric field or frequency increases beyond the saturation threshold, which in turn leads to a gradual increase in coercivity,^{1,2,3} as shown in Fig. S12a. The factors contributing to the increase in coercivity of ferroelectric materials are the intrinsic defects (e.g., vacancies) and extrinsic defects (e.g., heterointerfaces), which can act as pinning centers that anchor ferroelectric domain walls, thereby impeding their dynamic switching behavior.^{4,5,6} As illustrated in the experimental results, when the applied voltage was increased from 10 V to 12 V, the P-E loop exhibited a significant lateral broadening, while the saturation polarization remained nearly unchanged. This experimental observation is well-aligned with the characteristic behaviors of the defect pinning mechanism.

Supplementary Fig. S4 | EELS spectra showing the Ti $L_{2,3}$ edges in STO and SATO. The Ti $L_{2,3}$ edges of SATO and STO have very similar fine structures with four peaks, suggesting that the Ti ions in SATO have the valence state of +4.

Supplementary Table S1 | Quantitative statistics of EDS spectra in different areas of SATO films.

	Sr (Norm.at.%)	Al (Norm.at.%)	Ti (Norm.at.%)	O (Norm.at.%)	Sr : Al : Ti : O
SATO_EDS_#1	3.15	33.76	3.24	59.85	1.00 : 10.72 : 1.03 : 19.00
SATO_EDS_#2	3.12	33.35	3.24	60.29	1.00 : 10.69 : 1.04 : 19.32
SATO_EDS_#3	3.18	34.09	3.13	59.60	1.00 : 10.72 : 0.98 : 18.74

Reference

1. Balashova, E. *et al.* Structural properties and dielectric hysteresis of molecular organic ferroelectric grown from different solvents. *Crystals* 11, 1278 (2021).
2. Hao, P. *et al.* Highly enhanced polarization switching speed in HfO₂-based ferroelectric thin films via a composition gradient strategy. *Adv. Funct. Mater.* 33, (2023).
3. Shcherbakov A. *et al.* Evolution of coercive voltage in a ferroelectric memory cell based on Hf_{0.5}Zr_{0.5}O₂ during its lifetime. *Phys. Rev. Appl.* **22**, 064083 (2024).
4. Zhang, D. *et al.* Superior polarization retention through engineered domain wall pinning. *Nat. Commun.* 11, 349 (2020).
5. Gao, P. *et al.* Revealing the role of defects in ferroelectric switching with atomic resolution. *Nat. Commun.* **2**, 591 (2011).
6. Bencan, A. *et al.* Domain-wall pinning and defect ordering in BiFeO₃ probed on the atomic and nanoscale. *Nat. Commun.* 11, 1762 (2020).

Reply to referee 2 (R2):

We appreciate the general recognition by the referee that “The authors have made substantial improvements to the manuscript and addressed most of the concerns raised in the initial review. The additional experimental data, particularly, polarization-dependent photoresponse characterization, significantly strengthen the work”. We sincerely thank the referee’s comments and suggestions, which are very helpful for improving the manuscript.

Question and comment (R2.1): While the authors have provided PFM measurements of the AlN precursor film (Supplementary Fig. S9), this does not address the original concern about the pristine SATO film state. Required additional data is PFM amplitude and phase images of virgin SATO film regions (without any prior electrical poling).

The current response addresses whether the precursor AlN film is ferroelectric, but the fundamental question about the intrinsic domain state of the SATO film remains unanswered.

Reply to Question and comment (R2.1):

We sincerely appreciate your valuable suggestion and apologize for any misunderstanding regarding your original concern. We conducted PFM measurements on the SATO film in its initial state without any prior electrical poling, as shown in the Supplementary Fig. S9. Both the amplitude and phase images of the virgin SATO film exhibit uniform contrast, indicating that there are no ferroelectric domain walls in the observed region.

The following sentences and Supplementary Fig. S9 were added in the revised manuscript and Supplementary Information:

Supplementary Fig. S9 shows the PFM measurements of SATO film in its initial state without any prior electrical poling. Both the amplitude and phase images of the virgin SATO film exhibit uniform contrast, indicating that there are no ferroelectric domain walls in the observed region.

Supplementary Fig. S9 | PFM characterizations of the SATO thin films without any prior electrical poling. (a-c) PFM topography, amplitude, and phase images of the SATO thin film. Both the amplitude and phase images of the virgin SATO film exhibit uniform contrast, indicating that there are no ferroelectric domain walls in the observed region. Scale bar, 2 μm .

Question and comment (R2.2): The authors propose Al vacancies as the charge compensation mechanism for Ti^{4+} substitution. However, there appears to be an inconsistency between this explanation and the reported compositional analysis. The EDS analysis reports $\text{Sr}:\text{Al}:\text{Ti}:\text{O} = 1:11:1:19$, suggesting single Ti substitution. If the charge balance requires Al vacancies (as stated: "Al vacancies must exist in SATO"), the expected stoichiometry would differ from $\text{SrAl}_{11}\text{TiO}_{19}$. A more detailed explanation is needed for how the proposed defect chemistry (Al vacancies + Ti^{4+} substitution) reconciles with the experimental composition data.

Reply to Question and comment (R2.2):

We sincerely appreciate your valuable comments. According to the EELS spectra as shown in Supplementary Figure S5, the chemical valence state of Ti in the SATO thin film is +4. To achieve the electrical neutrality, Al vacancies must exist in the SATO thin film. Therefore, it is more reasonable to define the chemical formula of SATO as $\text{SrAl}_{11-\delta}\text{TiO}_{19}$, where δ represents a parameter related to Al vacancies. To determine more accurately the chemical composition in the SATO thin film, we performed EDS measurements in several different regions, as shown in Supplementary Table S1. After background subtraction, normalization, and integration of the EDS spectrum, the average atomic ratios of Sr, Al, Ti and O in the SATO film is 1.00:10.71:1.02:19.02. The atomic ratio of Sr to Ti is close to 1, similar as that in the SrTiO_3 substrate. Since the structure of SATO is derived from $\text{SrAl}_{12}\text{O}_{19}$ after one Ti atom replacing one Al atom, vacancies must be existed in the other eleven sites of Al. Since the valence states of Ti and Al ions are +4 and +3 respectively, the chemical composition of SATO should

be SrAl_{10.67}TiO₁₉ to reach the complete electrical neutrality ($\delta=0.33$ in SrAl_{11- δ} TiO₁₉), which is consistent with the EDS measurements considering the accuracy of this technique.

The following sentences and Supplementary Table S1 were added in the revised manuscript and Supplementary Information:

To maintain the charge neutrality, Al vacancies must exist in SATO. The chemical formula of SATO can be defined as SrAl_{11- δ} TiO₁₉, where δ represents a parameter related to Al vacancies. Energy dispersive X-ray spectroscopy (EDS) measurements were carried out, and the results are shown in in Supplementary Fig. S5 and Supplementary Table S1. After background subtraction, normalization, and integration of the EDS spectrum, the average atomic ratios of Sr, Al, Ti and O in the SATO film is 1.00:10.71:1.02:19.02. The atomic ratio of Sr to Ti is close to 1, similar as that in the SrTiO₃ substrate. Since the structure of SATO is derived from SrAl₁₂O₁₉ after one Ti atom replacing one Al atom, vacancies must be existed in the other eleven sites of Al. Since the valence states of Ti and Al ions are +4 and +3 respectively, the chemical composition of SATO should be SrAl_{10.67}TiO₁₉ to reach the complete electrical neutrality ($\delta=0.33$ in SrAl_{11- δ} TiO₁₉), which is consistent with the EDS measurements considering the accuracy of this technique.

Supplementary Table S1 | Quantitative statistics of EDS spectra in different areas of SATO films.

	Sr (Norm.at.%)	Al (Norm.at.%)	Ti (Norm.at.%)	O (Norm.at.%)	Sr : Al : Ti : O
SATO_EDS_#1	3.15	33.76	3.24	59.85	1.00 : 10.72 : 1.03 : 19.00
SATO_EDS_#2	3.12	33.35	3.24	60.29	1.00 : 10.69 : 1.04 : 19.32
SATO_EDS_#3	3.18	34.09	3.13	59.60	1.00 : 10.72 : 0.98 : 18.74

Reply to referee 3 (R3):

We appreciate the general recognition by the referee that “We thank the authors for their detailed and comprehensive reply. They provided additional information to clarify their claims. The paper is now much clearer and can be considered for publication.”

We sincerely thank all reviewers for their valuable comments, suggestions, and questions. Your comments have greatly helped improve this manuscript and encouraged us to pursue in-depth, comprehensive research on this topic. We hope our detailed responses to the above points address your concerns, and that the revised manuscript meets your requirements.

Reply to referees' questions and comments:

Manuscript ID: NCOMMS-25-34130B

Title: Ferroelectric ultraviolet photodetector material with ultrafast response speed

Authors: Xuexi Yan, Tingting Yan, Lingli Li, Yi Cao, Xinwei Wang, Jinghui Wang, Ang Tao, Tingting Yao, Yixiao Jiang, Weijin Hu, Xiaosheng Fang, Hengqiang Ye, Xiu-Liang Ma, Chunlin Chen

September 17, 2025

We would like to thank all reviewers for reviewing our manuscript very carefully and acknowledging our work. We fully understand the referee's concerns and here we address all the questions and discuss all the comments.

Reply to referee 1 (R1):

Thank you very much for your recognition that "The paper may be accepted for publication."

Reply to referee 2 (R2):

We appreciate the general recognition by the referee that "The authors have made substantial improvements to address most of the concerns raised in the initial review, and their efforts are acknowledged".

Question and comment (R2.1): The PFM measurements in Fig. S9 demonstrate that the as-prepared SATO sample exhibits characteristics consistent with a paraelectric phase. While the amplitude image is presented in arbitrary units (a.u.), preventing direct quantitative comparison with the results obtained under applied field, the phase image reveals distinctly different values compared to those observed after electrical poling (Fig. 4b, 4d). If the as-prepared sample were truly in a ferroelectric phase, the phase values should show consistency between the virgin state and post-poling measurements, at least within domains of the same polarization orientation.

The collective PFM results strongly suggest the possibility of a field-induced ferroelectric phase transition rather than intrinsic ferroelectricity. However, this interpretation directly contradicts the TEM structural analysis, which concludes that the as-prepared sample possesses a ferroelectric phase based on observed atomic displacements and structural characteristics.

These conflicting interpretations cannot be reconciled without further clarification and additional experimental evidence. Therefore, the manuscript cannot be considered for publication until this critical contradiction is satisfactorily resolved.

Reply to Question and comment (R2.1):

Thank you very much for your valuable comment. First, we sincerely apologize for the misunderstanding caused by the Fig. S9 in the former revision. Regarding to the ferroelectricity of the SATO film, our considerations are as follows:

HAADF STEM image in Fig. S13 shows clearly atomic displacements, which suggests that the as-prepared SATO film possesses a ferroelectric phase. TEM bright-field image in Fig. S7 shows that the SATO film has a uniform contrast. No ferroelectric domain walls can be observed. These results suggest that the SATO ferroelectric film has a single-domain structure in the observed region. Furthermore, the PFM phase image of the SATO film exhibits a uniform contrast, indicating that the observed region has the same ferroelectric polarization direction, which is consistent with the TEM observations. Thus, the ferroelectricity of the SATO film should be intrinsic, rather than a voltage-driven ferroelectric phase transition. Although the PFM phase image of a paraelectric material will also show a uniform contrast, the reason is different. The PFM phase image of the SATO film exhibits a uniform contrast because the whole observed region locates within a single ferroelectric domain of the same polarization direction.

Regarding the inconsistency between the phase values in Figure 4 and Figure S9 you mentioned, we carefully considered this question based on the working principle of PFM. For PFM testing, the essence of the phase image is that during the cantilever's vibration, its energy is dissipated or modulated by the sample, causing the vibration to not fully synchronize with the drive signal. The PFM device detects the phase lag of the forced vibration of the cantilever and converts the spatial distribution of the phase difference into a visual contrast image. The baseline of the PFM phase image is determined by the cantilever's initial phase. However, the initial phase of the cantilever is not a fixed value. The driving voltage, sample position, and probe type will affect the initial phase of the cantilever. Therefore, it is

difficult to make a direct quantitative comparison of the phase values obtained from two independent images. Thus, the phase values in Figure 4 and Figure S9 are different since they were obtained from two different measurements in which the initial phases of the cantilever were different. We cannot compare them directly.

To directly and quantitatively compare the phase changes of the SATO film under three conditions: no voltage, forward voltage, and reverse voltage, we re-performed a PFM test on the SATO film and showed the results in Figure S9 in the new revision. As one can see in Fig. S9c, the phase values in the 0 V region are almost identical, indicating that the SATO film initially has the same ferroelectric polarization direction. The phase values in the 0 V region and the -10V region are also identical, indicating that the initial direction of the ferroelectric polarization is perpendicular to the film surface and outward. Upon applying a +10V voltage, a 180° phase shift occurs, signifying a change in ferroelectric polarization direction from being perpendicular to the film surface and outward to an inward orientation. These results are consistent with the TEM structural analyses.

The following sentences and Supplementary Fig. S9 were added in the revised manuscript and Supplementary Information:

Supplementary Fig. S9 presents the PFM measurements of the SATO film under three conditions: no voltage, forward voltage, and reverse voltage. The amplitude and phase images of the virgin SATO film display uniform contrast, indicating the absence of ferroelectric domain walls in the observed region. The contrast between the 0 V region and the -10 V region remains consistent, suggesting that the initial ferroelectric polarization direction of the as-grown SATO film is perpendicular to its surface and directed outward. Upon applying a +10 V voltage, a 180° phase shift occurs, signifying a change in ferroelectric polarization direction from being perpendicular to the film surface and outward to an inward orientation.

Supplementary Fig. S9 | PFM characterizations of the SATO thin films. (a-c) PFM topography,

amplitude, and phase images of the SATO thin film. Both the amplitude and phase images of the SATO film exhibit a uniform contrast under 0 V and -10 V, indicating that the SATO is a single-domain ferroelectric film with the polarization along out-of-plane direction. Scale bar, 2 μm .

We hope the new experimental results and responses can address your concerns, and that the revised manuscript meets your requirements. Thank you very much for your valuable comments and precious suggestions, which have greatly helped improve this manuscript.

Reply to referees' questions and comments:

Manuscript ID: NCOMMS-25-34130C

Title: Ferroelectric ultraviolet photodetector material with ultrafast response speed

Authors: Xuexi Yan, Tingting Yan, Lingli Li, Yi Cao, Xinwei Wang, Jinghui Wang, Ang Tao, Tingting Yao, Yixiao Jiang, Weijin Hu, Xiaosheng Fang, Hengqiang Ye, Xiu-Liang Ma, Chunlin Chen

November 8, 2025

We are grateful to #2 reviewer for reviewing our manuscript carefully and recognizing our efforts. We systematically discussed the reviewer's concerns and conducted repeated measurements to confirm the reviewer's concerns about the reproducibility of the PFM results. We hope our response will reassure the reviewer and address their concerns.

Reply to referee 2 (R2):

We appreciate the general recognition by the referee that "The manuscript reports highly intriguing and potentially impactful experimental results regarding ultrafast ferroelectric photodetectors with exceptional performance characteristics. The main finding of nanosecond-scale response times combined with high responsivity and detectivity appears robust and could attract broad interest within the optoelectronics community."

Question and comment (R2.1): The manuscript reports highly intriguing and potentially impactful experimental results regarding ultrafast ferroelectric photodetectors with exceptional performance characteristics. The main finding of nanosecond-scale response times combined with high responsivity and detectivity appears robust and could attract broad interest within the optoelectronics community. However, the interpretation of these results relies on several supporting datasets, which unfortunately exhibit significant inconsistencies and lack reproducibility.

In particular, the additional PFM data provided in response to the review (Fig. S9) do not reconcile with the results in Fig. 4 and raise new concerns about reproducibility. In Fig. 4, PFM measurements demonstrate consistent piezoelectric amplitude regardless of polarization direction upon ± 10 V application, supporting the claimed single-domain behavior. However, in Fig. S9, despite nearly

identical measurement conditions, the region subjected to -10 V application exhibits a distinctly larger response than the region subjected to $+10$ V. Furthermore, although the application of -10 V from the unpoled state does not induce any change in the polarization direction, an increase in the piezoelectric response is observed, which is in clear contradiction with the single-domain nature claimed on the basis of the TEM observations.

The observed photodetection properties themselves appear genuine and noteworthy. However, the proposed ferroelectric mechanism relies on materials characterization that contains contradictory evidence and lacks internal consistency. Without a clear and consistent experimental foundation, the mechanistic explanation for the exceptional photodetector performance remains insufficiently supported. While the authors have made efforts to address previous concerns, the newly provided data have inadvertently highlighted fundamental issues with experimental reproducibility. In the current form, the manuscript does not meet the standard for Nature Communications.

Reply to Question and comment (R2.1):

Thank you very much for your comments. However, we cannot agree with your comments on PFM measurements and experimental reproducibility. Let's return to the original comment in the 1st-round review: to prove that the as-grown SATO ferroelectric thin film exhibits a single-domain characteristics. First, to determine the initial ferroelectric domain state of the as-grown thin film, it is necessary to avoid any voltage interference since the polarization direction of ferroelectric domains are easily affected by applied voltage. Therefore, we believe that your conclusion about "In Fig. 4, PFM measurements demonstrate consistent piezoelectric amplitude regardless of polarization direction upon ± 10 V application, supporting the claimed single-domain behavior" is not rigorous, because a positive or negative voltage was applied to the observation area in Figure 4. Amplitude images of PFM are not sensitive to the polarization direction, the single-domain nature can be supported by phase images, instead of amplitude images. Second, surface charge injection and surface potential changes will significantly affect the amplitude. Extensive studies have reported that the surface charge injection and potential changes are common and almost unavoidable during PFM measurements (*Nat. Mater.* 2022, 21, 909-909; *Nat. Commun.* 2016, 7, 11502; *Nat. Commun.* 2017, 8, 15217; *J. Semicon.* 2019, 40:9, 092002), which can clarify your concern on "Furthermore, although the application of -10 V from the unpoled state does not induce any change in the polarization direction, an increase in the piezoelectric response is observed". Third, surface potential varies with applied voltages, leading to

the differences in amplitude under negative and positive voltages with the same value (*ACS Appl. Electron. Mater.* 2024, 6, 6401–6410; *Appl. Phys. Rev.* 2017, 4, 021302; *ACS Nano* 2014, 10, 10223–10236), which can explain your concern about “However, in Fig. S9, despite nearly identical measurement conditions, the region subjected to –10 V application exhibits a distinctly larger response than the region subjected to +10 V.” Phase images in PFM measurements are highly sensitive to polarization direction. Therefore, if we want to determine the initial ferroelectric domain state of the as-grown thin film, we should focus more on identifying phase information (*Adv. Funct. Mater.* 2025, 35, 2423225; *Nat. Mater.* 2022, 21, 909–909; *Nat. Commun.* 2017, 8, 15217). Amplitude images are typically used to identify the locations of domain walls since amplitude value often abruptly changes at domain walls. The amplitude images in this study are uniform, suggesting that there are no domain walls inside the measured regions, which is also consistent with the single-domain nature.

In summary, we believe there are two aspects to provide direct evidence for accurately proving the initial domain state and polarization direction of SATO thin film: **i) obtaining a PFM phase image without applied voltage, which can determine whether the film is a single-domain film; ii) applying positive and negative voltages polarization writing and comparing the phase images with and without applied voltages, which can determine the polarization direction.** Considering your concerns about the reproducibility of our PFM measurements, we take this very seriously and have immediately repeated the PFM results (shown in Supplementary Figure S9) and we are confident that the PFM measurements of the SATO thin films have good reproducibility. As shown in Figure S9, we measured four different positions on a $10 \times 10 \times 0.5$ mm sample under nearly identical measurement conditions. We found that the initial amplitude and phase of all the positions had uniform contrast, which directly indicated that the as-grown SATO film uniform and has an intrinsic single-domain structure. Subsequently, we wrote polarization patterns at the same positions and found that the regions with applied -10 V had similar phase contrast to the regions without applied voltage. This indicates that the initial direction of the ferroelectric polarization is perpendicular to the film surface and outward, which is consistent with our STEM HAADF image shown in Supplementary Figure S14. Upon applying a +10V voltage, a 180° phase shift occurs, signifying a change in ferroelectric polarization direction from being perpendicular to the film surface and outward to an inward orientation. Therefore, based on the initial amplitude and phase images of the PFM, we can prove that the as-grown SATO film exhibits the single-domain nature. The polarization patterns written with

positive and negative voltages prove that SATO possesses the intrinsic ferroelectricity.

The following sentences and Supplementary Fig. S9 of the new version were added in the revised manuscript and Supplementary Information:

To determine the initial domain state of the virgin SATO film and verify the reproducibility of the PFM measurements, we performed multiple measurements at four randomly selected positions on a 10×10 mm sample under nearly identical measurement conditions, as shown in Supplementary Fig. S9. The initial piezoelectric amplitude and phase images at all four positions display uniform contrast, which indicates that the virgin SATO film exhibits the single-domain nature. The phase contrast between the 0 V region and the -10 V region remains almost the same, suggesting that the initial ferroelectric polarization direction of the as-grown SATO film is perpendicular to its surface and directed outward. Furthermore, a +10 V writing voltage induces a 180° phase changes, suggesting the flip of the polarization direction from upward to downward. The PFM measurements at all positions showed similar characteristics, indicating that the SATO film is uniform with a good reproducibility of ferroelectric properties.

Thank you again for your careful review. We sincerely hope the new experimental results and responses can address your concerns and meet your requirements. Thank you very much for your valuable comments, which have greatly helped improve our manuscript.

Supplementary Fig. S9 | PFM characterizations of the SATO thin films. (a-d) PFM topography, amplitude, and phase images over $10 \times 10 \mu\text{m}^2$ at four different positions. The inset in **a** shows the measurement positions. Both the initial amplitude and phase images of the SATO film exhibit a uniform contrast under 0 V and -10 V, indicating that the SATO is a single-domain ferroelectric film with the polarization along out-of-plane direction. Scale bar, 2 μm .

Reply to referees' questions and comments:

Manuscript ID: NCOMMS-25-34130D

Title: Ferroelectric ultraviolet photodetector material with ultrafast response speed

Authors: Xuexi Yan, Tingting Yan, Lingli Li, Yi Cao, Xinwei Wang, Jinghui Wang, Ang Tao, Tingting Yao, Yixiao Jiang, Weijin Hu, Xiaosheng Fang, Hengqiang Ye, Xiu-Liang Ma, Chunlin Chen

November 27, 2025

We are grateful to #2 reviewer for reviewing our manuscript carefully and recognizing our efforts. We have fully incorporated the reviewer's suggestions and revised the manuscript to improve its logical consistency.

Reply to referee 2 (R2):

We appreciate the recognition by the referee that "The reviewer appreciates the additional PFM data (Fig. S9). These PFM images demonstrate that the PFM measurements for the pristine regions exhibit a certain degree of reproducibility, which strengthens the basis for discussion."

Question and comment (R2.1):

The reviewer appreciates the additional PFM data (Fig. S9).

These PFM images demonstrate that the PFM measurements for the pristine regions exhibit a certain degree of reproducibility, which strengthens the basis for discussion.

Based on the revised manuscript, I would like to summarize once again the evidence supporting the ferroelectricity of the SATO thin films:

Observation of the displacement of Al atoms by HAADF

Clear polarization switching behavior in the PUND measurements

Photoresponse characteristics dependent on polarization direction

These results support the conclusion that SATO exhibits ferroelectricity.

On the other hand, the evidence for the single-domain nature of SATO requires more careful evaluation.

The authors clearly acknowledge that surface charge injection and changes in surface potential significantly influence the PFM amplitude. Indeed, the results in Fig. S9 show that the SATO film is

strongly affected by surface charge injection—an effect much larger than the intrinsic piezoelectric response. Therefore, the possibility of charge-injection-induced contrast cannot be fully excluded, especially because an AC electric field is applied even in the pristine regions during PFM measurements.

Thus, while PFM results may serve as auxiliary information, they should not be treated as conclusive evidence. The description of a single-domain state should be revised to a more modest expression, such as “indicating the possibility of a single domain.”

Furthermore, the presence of surface charge injection also affects the interpretation of the PFM data in Fig. 4. The polarization direction can be meaningfully discussed based on phase only when the piezoelectric response dominates over the surface-charge-related contribution. However, because the SATO film is strongly affected by charge injection, the phase contrast in Fig. 4 should be removed.

I strongly recommend that the primary evidence for ferroelectricity should rely on PUND measurements and structural analysis.

If these revisions are made, the logical consistency of the paper will be greatly improved, and I believe it will reach a level worthy of publication in Nature Communications.

Reply to Question and comment (R2.1):

Thank you very much for your valuable comments and suggestions. According to your precious suggestions, the description of a single-domain state was revised to a more modest expression. We revised the description of "single-domain state" to "possibility of a single domain" in the whole manuscript, which is indeed the most comprehensive and prudent expression.

We agree that the primary evidence for ferroelectricity should rely on PUND measurements and structural analysis. Thus, we rearranged the figures and removed the phase contrast in Fig. 4, and added the ferroelectric fatigue properties tested by PUND measurements in Figs. 4d,e which were previously shown in Supplementary Figures. The phase contrast in Fig. 4 was moved to Supplementary Fig. S10 to demonstrate that the SATO film exhibits excellent retention after poling.

The revised sentences and Fig. 4 are as following.

“...exhibits the **possibility** of single-domain ferroelectrics”

“...indicates that the SATO ferroelectric film **probably** has a single-domain structure”

“...the virgin SATO film **possibly possesses** single-domain nature”.

Fig. 4 | Ferroelectric and piezoelectric properties of the SATO thin film. (a) PFM topographic image of the SATO thin film. The RA is about 2 nm. **(b)** PFM amplitude image polarized by ± 10 V bias voltages. The voltage of the probe tip (V_{tip}) was 2 V. **(c)** The local PFM amplitude (red) and phase (blue) hysteresis loops. **(d)** The polarization-electric field (P-E) hysteresis loops of the SATO thin film under different voltages at 300 K, 1 kHz. The SATO thin film has a remanent polarization of $7.8 \mu\text{C}/\text{cm}^2$ and exhibits typical intrinsic ferroelectric characteristics. **(e)** Fatigue characteristics of the SATO thin film under 10 V and 12 V. Insets are the PUND loops before and after cycling. After 10^6 cycles, the polarization intensity exhibits a strong fatigue stability without significant attenuation. Scale bar, $2 \mu\text{m}$.

Sincerely thank you again for your careful review and precious suggestions. We hope the responses can address your concerns and meet your requirements. Thank you very much for your valuable comments and suggestions, which have greatly helped improve our manuscript.